# MODALITY-FREE GRAPH IN-CONTEXT ALIGNMENT

**Wei Zhuo, Siqiang Luo**
Nanyang Technological University
{wei.zhuo, siqiang.luo}@ntu.edu.sg

## ABSTRACT

In-context learning (ICL) converts static encoders into task-conditioned reasoners, enabling adaptation to new data from just a few examples without updating pretrained parameters. This capability is essential for graph foundation models (GFMs) to approach LLM-level generality. Yet current GFMs struggle with cross-domain alignment, typically relying on modality-specific encoders that fail when graphs are pre-vectorized or raw data is inaccessible. In this paper, we introduce Modality-Free Graph In-context Alignment (MF-GIA), a framework that makes a pretrained graph encoder promptable for few-shot prediction across heterogeneous domains without modality assumptions. MF-GIA captures domain characteristics through gradient fingerprints, which parameterize lightweight transformations that align pre-encoded features and indexed labels into unified semantic spaces. During pretraining, a dual prompt-aware attention mechanism with episodic objective learns to match queries against aligned support examples to establish prompt-based reasoning capabilities. At inference, MF-GIA performs parameter-update-free adaptation using only a few-shot support set to trigger cross-domain alignment and enable immediate prediction on unseen domains. Experiments demonstrate that MF-GIA achieves superior few-shot performance across diverse graph domains and strong generalization to unseen domains. The code is available at https://github.com/JhuoW/MF-GIA.

## 1 INTRODUCTION

The remarkable success of Large Language Models (LLMs) has fundamentally revolutionized AI, with in-context learning (Brown et al., 2020; Zhang et al., 2023; Lu et al., 2022) emerging as a pivotal capability that enables these models to adapt to new tasks through mere exposure to a few demonstration examples, without any parameter updates like fine-tuning. This paradigm shift, from task-specific fine-tuning to prompt-based adaptation, naturally sparks a profound question for the graph learning com-

Table 1: Comparison of methods with respect to the three main criteria of true ICL.

| Method | Post-Training Free | Domain Alignment | Modality-Free |
|---|:---:|:---:|:---:|
| SSL-GNN | ✗ | ✗ | ✓ |
| All in One (Sun et al., 2023) | ✗ | ✗ | ✓ |
| GPF (Fang et al., 2023) | ✓ | ✗ | ✓ |
| GCOPE (Zhao et al., 2024) | ✗ | ✓ | ✓ |
| GFT (Wang et al., 2024b) | ✗ | ✓ | ✗ |
| Prodigy (Huang et al., 2023) | ✓ | ✗ | ✓ |
| Unigraph (He et al., 2025a) | ✓ | ✓ | ✗ |
| AutoGFM (Chen et al., 2025) | ✗ | ✓ | ✗ |
| GraphAlign (Hou et al., 2024) | ✓ | ✓ | ✗ |
| OFA (Liu et al., 2024a) | ✓ | ✓ | ✗ |
| GOFA (Kong et al., 2025) | ✓ | ✓ | ✗ |
| **MF-GIA** | ✓ | ✓ | ✓ |

munity: *Can we achieve similar foundation-level generality for graph-structured data?* Unlike sequential text where context flows naturally, graphs encode complex topological patterns, multi-hop dependencies, and heterogeneous node and edge attributions that demand fundamentally new approaches to demonstration selection, prompt design, and reasoning.

Achieving true graph in-context learning demands three fundamental criteria that remain elusive in existing methods. First, **post-training-free** inference is essential for genuine ICL, where models must adapt to new tasks through demonstrations alone, without fine-tuning or learnable prompt engineering. Second, **cross-domain alignment** enables a single model to reason across diverse graph types within a unified semantic space, mirroring LLMs' domain-agnostic capabilities. Third, **modality-free** operation ensures that the model can process arbitrary pre-encoded graphs without requesting raw data, crucial for the heterogeneous domains of real-world graphs. As shown in Table 1, prior approaches fall short of meeting all three criteria at once: self-supervised GNNs (You et al., 2020; Qiu et al., 2020) and GFMs like All in One (Sun et al., 2023), GCOPE (Zhao et al.,

2024), and GFT (Wang et al., 2024b) compromise ICL through required post-training on downstream graphs, where All in One trains task-specific prompts while GCOPE and GFT require fine-tuning; GPF (Fang et al., 2023) and Prodigy (Huang et al., 2023) lack cross-domain alignment, limiting their generalization on graphs from unseen domains; recent advances like UniGraph (He et al., 2025a), GraphAlign (Hou et al., 2024), and OFA (Liu et al., 2024a) achieve alignment and post-training-free inference, yet sacrifice modality freedom by requiring conversion to a single unified modality (e.g., text-attributed graphs) for alignment, making them inapplicable when raw data are inaccessible or when graphs are already pre-encoded by domain-specific pipelines. More related work is discussed in Appendix A.

In this work, we present Modality-free Graph In-context Alignment (MF-GIA), the first GFM to achieve all three criteria for true in-context learning on graphs. Our key insight is that the interaction between a graph and a shared frozen encoder reveals its domain characteristics, which can be captured by a gradient fingerprint: a single-step parameter update that encodes how features, labels, and structure jointly influence the model. This fingerprint drives lightweight domain-conditioned transformations that align pre-encoded features and graph-local label IDs into unified semantic spaces, where related domains occupy neighboring subspaces while preserving intra-domain geometry, thereby achieving modality-free domain alignment. The aligned features and labels are then processed by Dual Prompt-Aware Attention (DPAA) optimized with an episodic objective that learns to match queries against support examples. This approach establishes prompt-based in-context reasoning that simulates the few-shot scenarios faced at test time. At inference, given a few labeled examples as prompts, MF-GIA computes the fingerprint, instantiates the aligners, and performs parameter-update-free prediction on unseen domains. Experiments across diverse benchmarks demonstrate that MF-GIA excels at few-shot node-level tasks, generalizes to entirely unseen domains without additional training, and transfers seamlessly to edge-level tasks. These results bring GFMs closer to the universal in-context learning capabilities exhibited by LLMs.

## 2 PRELIMINARIES

Following the ICL setup of the pioneering work Prodigy (Huang et al., 2023), we study few-shot, prompt-based node and edge classification. In this section, we formalize graph ICL as episodic classification over graphs and introduce a modality-free alignment perspective that standardizes features and labels across domains.

### 2.1 GRAPH IN-CONTEXT LEARNING

Let $\mathcal{G} = \{G_1, G_2, \cdots, G_M\}$ denote a collection of $M$ graphs drawn from heterogeneous domains, where each graph $G_i = (V_i, E_i, \mathbf{X}_i, \mathbf{Y}_i)$ comprises a node set $V_i$, an edge set $E_i \subseteq V_i \times V_i$, node features $\mathbf{X}_i = \{x_{i,1}, \cdots, x_{i,|V_i|}\} \in \mathbb{R}^{|V_i| \times d_i}$, and labels $\mathbf{Y}_i$. The node features $\mathbf{X}_i$ may exist in domain-specific formats (e.g., dense vectors, categorical attributes, IDs), with potentially different dimensions $d_i$. To pretrain a universal model across these graphs with a common input width, following (Yu et al., 2025; Zhao et al., 2024), we first unify feature dimensions to $d_o$ by applying SVD on each $\mathbf{X}_i \leftarrow \text{SVD}(\mathbf{X}_i) \in \mathbb{R}^{|V_i| \times d_o}$. Depending on the task, $\mathbf{Y}_i$ is either a node-label vector $\mathbf{Y}_i^{\text{node}} = \{0, 1, \cdots, C_i^{\text{node}} - 1\}^{|V_i|}$ or an edge-label vector $\mathbf{Y}_i^{\text{edge}} = \{0, 1, \cdots, C_i^{\text{edge}} - 1\}^{|E_i|}$, where $C_i^{\text{node}}$ and $C_i^{\text{edge}}$ denote the number of node and edge classes in $G_i$, respectively. A universal GNN encoder $f_\theta : \mathbb{R}^{d_o} \to \mathbb{R}^d$ maps graph items (nodes or edges) to $d$-dimensional representations. For node classification, the representation of $v \in V_i$ is $h_{i,v} = f_\theta(v; G_i) \in \mathbb{R}^d$. For edge classification, we analogously obtain $\mathbf{h}_{i,e} = f_\theta(e; G_i)$ for $e \in E_i$ using endpoint features and structure as needed. We use the generic symbol $w$ to denote an item ($w = v$ or $w = e$).

Given $\mathcal{G}$ as a pretraining corpus with $M$ graphs and a target graph $G_{\text{new}} = (V_{\text{new}}, E_{\text{new}}, \mathbf{X}_{\text{new}}, \mathbf{Y}_{\text{new}})$ from an unseen domain with $C_{\text{new}}$ classes, *graph in-context learning* aims to classify graph items in $G_{\text{new}}$ using a few labeled examples per class as in-context demonstrations, without updating model parameters. Formally, the graph ICL operates in two phases. During pretraining, we learn a unified model $\mathcal{M}_\Phi : \mathcal{G} \to \mathcal{Y}$ on the corpus $\mathcal{G}$. At test time, given a support set $\mathcal{S} = \{(w_j, y_j)\}_{j=1}^{k \cdot C_{\text{new}}}$ containing $k$ labeled graph items per class from $G_{\text{new}}$ as prompts, the model predicts labels for query items $\mathcal{Q} = \{q : q \in G_{\text{new}} \backslash \mathcal{S}\}$ as:

$$\hat{y}_q = \mathcal{M}_\Phi(q, G_{\text{new}}, \mathcal{S}), \quad \forall q \in \mathcal{Q}, \tag{1}$$

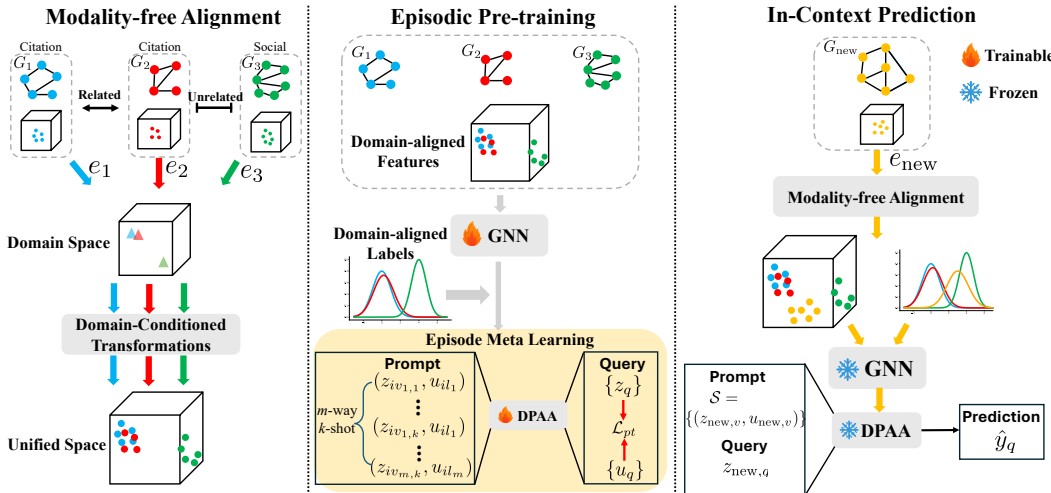

Figure 1: Overview of MF-GIA. **(Left) Modality-free Alignment:** The pretraining graphs are mapped to a unified space via domain-conditioned transformations. Domain descriptors $e$ ensure similar domains occupy neighboring subspaces. **(Middle) Episodic Pretraining:** The model learns from $m$-way $k$-shot episodes using domain-aligned features and labels. The DPAA mechanism matches queries to classes using only prompts as context. **(Right) In-context Prediction:** For an unseen graph, the frozen model performs few-shot classification using the support set as a prompt.

where the pretrained model $\mathcal{M}_\Phi$ is parameterized by $\Phi$. Crucially, $\Phi$ remains frozen during inference, so the model leverages the in-context demonstrations in $\mathcal{S}$ to adapt to the new domain without fine-tuning. For example, consider $\mathcal{M}_\Phi$ pretrained on citation and E-commerce networks. When tested on a social network $G_{\text{new}}$ from an unseen domain, the model can classify users in $G_{\text{new}}$ without fine-tuning. Instead, we provide a support set containing a few labeled users from each class. By leveraging these in-context demonstrations as prompts, $\mathcal{M}_\Phi$ identifies patterns between the support examples and query users to classify the remaining users, all while keeping its parameters frozen.

**Episodic Meta Learning.** To enable in-context adaptation, we adopt an episodic training paradigm (Vinyals et al., 2016; Li et al., 2019). Specifically, for each pretraining graph $G_i \in \mathcal{G}$, we construct $m$-way $k$-shot episodes by sampling $m$ classes and $k$ examples per class as a support set $\mathcal{S}$, with additional samples as queries $\mathcal{Q}$. The model $\mathcal{M}_\Phi$ consumes $(G_i, \mathcal{S})$ as the prompt and is optimized to maximize the likelihood of the ground-truth labels on $\mathcal{Q}$:

$$\min_\Phi \mathbb{E}[-\frac{1}{|\mathcal{Q}|} \sum_{q \in \mathcal{Q}} \log \hat{p}(y_q \mid q, \mathcal{S}, G_i)]. \tag{2}$$

This episodic formulation teaches the model to recognize patterns from limited examples. By pretraining on numerous episodes that simulate the few-shot scenarios encountered at test time, the model acquires the capacity to perform in-context reasoning. At inference, this enables adaptation to new domains through few-shot prompts alone, with all pretrained parameters $\Phi$ remaining frozen.

## 2.2 MODALITY-FREE ALIGNMENT

The pretraining graphs in $\mathcal{G}$ often differ in both input modalities and label systems. Features range from dense vectors to categorical attributes or domain-specific identifiers. Likewise, label spaces are graph-local and vary in cardinality, with no global alignment across domains. These heterogeneities make direct in-context transfer across graphs challenging.

Recent GFMs (Wang et al., 2024b; He et al., 2025a; Liu et al., 2024a) attempt to unify graphs from heterogeneous domains with Text-Attributed Graphs (TAGs), which convert all features and labels to natural language, then map them with language models into shared semantic spaces. However, this approach has fundamental limitations. Real-world graph data is typically already vectorized through domain-specific methods, such as word2vec (Mikolov et al., 2013) for documents, molecular

fingerprints (Rogers & Hahn, 2010) for compounds, and user behavior embeddings (Pan & Ding, 2019). Converting these optimized representations to text and back introduces information loss and computational overhead. Furthermore, privacy constraints often restrict access to raw data, and data providers usually release only pre-encoded datasets, making modality-aware conversions infeasible in sensitive domains. Instead, we adopt a *modality-free alignment* perspective, which aligns graphs directly in their existing representations without modality-aware conversion.

**Definition 1.** *(Modality-free Alignment) Let $\{(G_i, L_i)\}_{i=1}^M$ be graphs from $M$ domains, whose item features are already pre-encoded by (unknown) domain-specific pipelines, $\mathbf{X}_i \in \mathbb{R}^{d_o}$, and whose labels have been indexed by $L_i = \{0, \cdots, C_i - 1\}$. A modality-free alignment is a domain-conditioned transformation system $\mathcal{T} = \left\{ (\mathcal{K}_i^{feat}, \mathcal{K}_i^{label}) \right\}_{i=1}^M$ with:*

$$\mathcal{K}_i^{feat} : \mathbb{R}^{d_o} \to \mathbb{R}^d \quad \text{(feature alignment)} \quad and \quad \mathcal{K}_i^{label} : L_i \to \mathbb{R}^d \quad \text{(label alignment)} \tag{3}$$

*that maps domain-specific features and label IDs directly into a unified $d$-dimensional feature space and label space, respectively, without reconstructing or converting to any intermediate modality. The transformations should be conditioned on the domain descriptors $e_i \in \mathbb{R}^{d_e}$ that capture domain characteristics of $G_i$, such that for any two domain $i, j$,*

$$\|\mathcal{K}_i^{feat} - \mathcal{K}_j^{feat}\| \propto \|e_i - e_j\| \quad and \quad \|\mathcal{K}_i^{label} - \mathcal{K}_j^{label}\| \propto \|e_i - e_j\|. \tag{4}$$

*This ensures that similar domains with close descriptors $e_i \approx e_j$ produce similar transformations, causing their aligned features and labels to occupy neighboring subspaces in the unified space.*

Fig. 1-left illustrates the idea intuitively. Modality-free alignment maps every graph into a unified semantic space according to domain relationships: graphs from related domains (e.g., two citation networks $G_1$ and $G_2$) have similar domain descriptors and thus map to neighboring subspaces, whereas unrelated domains (social network $G_3$) sit far away. The domain-conditioned transformations $\mathcal{K}_i^{\text{feat}}$ and $\mathcal{K}_i^{\text{label}}$ project each graph's pre-encoded features and indexed labels into unified feature and label spaces, preserving intra-domain semantics while enabling cross-domain transfer. This is essential because numerically similar feature vectors from different domains can carry entirely different meanings (each domain's encoder defines its own coordinate system), and indexed label IDs $[0, 1, 2, \cdots]$ are reused with domain-specific semantics. Modality-free alignment reconciles these differences by calibrating features and labels via the domain descriptor, unifying them in shared spaces without requiring any knowledge of the original data modality like TAGs.

## 3 MF-GIA: MODALITY-FREE GRAPH IN-CONTEXT ALIGNMENT

In this section, we present MF-GIA for enabling in-context learning across heterogeneous graph domains without modality-specific priors. MF-GIA addresses the fundamental challenge of aligning graphs with incompatible feature spaces and label systems through three key components: (1) domain embedder encodes domain characteristics, (2) domain-conditioned alignment maps pre-encoded features and indexed labels to unified spaces, and (3) episodic pretraining realizes few-shot adaptation during pretraining. We then describe in-context inference on graphs from unseen domains.

### 3.1 DOMAIN EMBEDDER

Reliable domain embeddings are the pivot of MF-GIA: they summarize graphs' domain characteristics, parameterize the domain-conditioned aligners $(\mathcal{K}_i^{\text{feat}}, \mathcal{K}_i^{\text{label}})$, and ensure that graphs with related domains are mapped to neighboring subspaces while preserving intra-/cross- domain semantics. Prior work represents graph domains using learnable tokens, but depends on external signals,

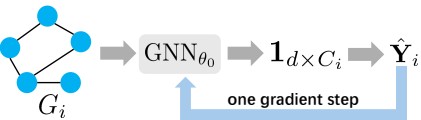

Figure 2: Domain embedder.

such as domain labels (Yu et al., 2025) or modality metadata (He et al., 2025a), which are often unavailable in practice. We instead induce the domain embeddings $\{e_i\}_{i=1}^M$ directly from each graph's intrinsic properties by capturing the interactions between the graph and a shared encoder, without external signals. Starting from a single shared weight initialization $\theta_0 \in \mathbb{R}^{d_o \times d}$ for a one-layer GNN encoder followed by a fixed all-ones projection matrix $\mathbf{1}_{d \times C_i}$ from embedding to label space as shown

in Fig. 2, we take a single gradient step on each pretraining graph $G_i \in \mathcal{G}$ as $\theta_i = \theta_0 - \eta \nabla_\theta \mathcal{L}_i (\theta_0)$, where $\eta$ is a small learning rate uniformly set to 0.01 and $\mathcal{L}_i$ is the task loss w.r.t. the available labels on $G_i$. The resulting single-step displacement $\Delta \theta_i = \theta_i - \theta_0$ serves as a *gradient fingerprint* that captures **how the graph's features, labels, and structure jointly influence the shared encoder**. Intuitively, graphs with similar gradient patterns are likely to come from related domains, making $\Delta \theta_i$ a natural descriptor of domain-level information. To obtain compact domain embeddings, we project these fingerprints through a learnable domain embedder $f_{\phi_{\text{de}}} : \mathbb{R}^{d_o \times d} \to \mathbb{R}^{d_e}$:

$$e_i = f_{\phi_{\text{de}}} (\Delta \theta_i) = \text{MLP} \left( \text{Flatten} \left( \text{Conv2D} \left( \Delta \theta_i \right) \right) \right) \in \mathbb{R}^{d_e}, \tag{5}$$

where the fingerprint $\Delta \theta_i \in \mathbb{R}^{d_o \times d}$ is treated as a single-channel image to be embedded. The embedder $f_{\phi_{\text{de}}}$ is trained to preserve domain relationships by minimizing:

$$\mathcal{L}_{\text{de}} = \sum_{G_i, G_j \in \mathcal{G}} \left( \left\| \Delta \theta_i - \Delta \theta_j \right\|_F - \left\| e_i - e_j \right\|_2 \right)^2, \tag{6}$$

so that pairwise relationships among graphs are retained in the embedding space. This approach naturally captures domain characteristics without domain labels or modality priors, as the gradient pattern inherently reflects the unique way each domain's data distribution interacts with the shared model initialization.

The domain embedding induced by the gradient fingerprint is central to MF-GIA, and the subsequent alignment operations are established on it. To justify its effectiveness, we provide a theoretical analysis showing that this embedding faithfully preserves domain characteristics (Proof in Appendix B.1).

**Theorem 3.1.** *Let $G_i$ and $G_j$ be graphs sampled from domains $\mathcal{D}_i$ and $\mathcal{D}_j$ respectively, with corresponding gradient fingerprints $\Delta \theta_i, \Delta \theta_j \in \mathbb{R}^{d_o \times d}$ computed using task loss $\mathcal{L}_i$ and $\mathcal{L}_j$ (e.g., cross-entropy). The domain embedder $f_{\phi_{de}}$ produces domain embeddings $e_i$ and $e_j$. Assuming every task loss $\mathcal{L}$ is $\mathscr{L}_{task}$-smooth with respect to model parameters, and $f_{\phi_{de}}$ has Lipschitz constant $\mathscr{L}_{de}$, the domain embeddings preserve domain relationships:*

$$\left\| e_i - e_j \right\|_2 \leq \widetilde{C} \cdot \mathcal{W}_2 \left( \mathcal{D}_i, \mathcal{D}_j \right) \tag{7}$$

*where $\mathcal{W}_2(\cdot, \cdot)$ measures inherent distance between two domains, and $\widetilde{C}$ is a constant.*

This upper bound ensures that if two domains are inherently similar, their embeddings learned by $f_{\phi_{\text{de}}}$ will be close in the embedding space, while dissimilar domains produce distant embeddings.

**In-context Domain Embedding.** For a downstream graph $G_{\text{new}}$ from an unseen domain, we compute its domain embedding using the same fingerprinting process. Given a few labeled items $\mathcal{S} = \{(w_i, y_i)\}_{i=1}^{k \cdot C_{\text{new}}}$ as a $C_{\text{new}}$-way $k$-shot prompt from $G_{\text{new}}$, we perform a single gradient step from the same initialization $\theta_0$, which is a component of the pretraining model $\mathcal{M}$ ($\theta_0 \in \Phi$), as $\theta_{\text{new}} = \theta_0 - \eta \nabla_\theta \mathcal{L}_{\text{new}}(\theta_0, \mathcal{S})$, where $\mathcal{L}_{\text{new}}$ is computed using only the prompt $\mathcal{S}$. The in-context domain embedding of $G_{\text{new}}$ is then computed by passing the gradient fingerprint through the pretrained domain embedder:

$$e_{\text{new}} = f_{\phi_{\text{de}}} \left( \theta_{\text{new}} - \theta_0 \right). \tag{8}$$

This process automatically captures $G_{\text{new}}$'s characteristics and positions it within the learned domain space. Since the domain embedder $f_{\phi_{\text{de}}}$ has been trained to preserve domain relationships during pretraining, it naturally maps the new graph's fingerprint to an appropriate location based on the knowledge learned from existing domains.

## 3.2 DOMAIN-CONDITIONED ALIGNMENT

With the domain embedding $e_i$ for $G_i \in \mathcal{G}$, MF-GIA instantiates two lightweight transformations $(\mathcal{K}_i^{\text{feat}}, \mathcal{K}_i^{\text{label}})$ as aligners that respectively align $G_i$'s item features and graph-local label IDs into unified semantic spaces. Because the transformations are conditioned on $e_i$, related domains with nearby $e_i$ induce similar transformations and occupy neighboring subspaces after alignment, while dissimilar domains remain separated. Here, we detail the feature and label alignment mechanisms and their application during in-context inference.

### 3.2.1 FEATURE ALIGNMENT

For each pretraining graph $G_i$, we learn a domain-conditioned feature transformation $\mathcal{K}_i^{\text{feat}}$ mapping pre-encoded item features to a unified feature space. Given an item $w \in G_i$ with its feature $x_w \in \mathbb{R}^{d_o}$, we first obtain its base representation via a shared GNN encoder $f_\theta$, whose first-layer weight matrix is initialized from the stored $\theta_0$:

$$h_{i,w} = f_\theta(w, G_i) \in \mathbb{R}^d. \tag{9}$$

Then we apply Feature-wise Linear Modulation (FiLM) (Perez et al., 2018) to generate domain-conditioned transformations from the domain embedding:

$$
\begin{aligned}
\left(\gamma_i^{\text{feat}}, \beta_i^{\text{feat}}\right) &= f_{\phi_{\text{feat}}}(e_i), \quad \gamma_i^{\text{feat}}, \beta_i^{\text{feat}} \in \mathbb{R}^d, \\
z_{i,w} &= \mathcal{K}_i^{\text{feat}}(h_{i,w}) = \gamma_i^{\text{feat}} \odot h_{i,w} + \beta_i^{\text{feat}},
\end{aligned}
\tag{10}
$$

where $f_{\phi_{\text{feat}}} : \mathbb{R}^{d_e} \to \mathbb{R}^{2d}$ is a two-layer MLP that outputs scale $\gamma_i^{\text{feat}}$ (with SoftPlus head for positivity) and shift $\beta_i^{\text{feat}}$ parameters, $\odot$ denotes element-wise product, and $z_{i,w}$ is the aligned feature for $w$ in $G_i$. The FiLM-based transformation $\mathcal{K}_i^{\text{feat}}$ is affine, so it calibrates scales and offset across domains to map features to a domain-specific subspace determined by $e_i$, while preserving the intra-domain geometry already present in $h_{i,w}$. Formally, the alignment satisfies (Proof in Appendix B.2):

**Property 1.** *If $f_{\phi_{\text{feat}}}$ is $\mathscr{L}$-Lipschitz continuous, then $\|\gamma_i^{\text{feat}} - \gamma_j^{\text{feat}}\|_2 + \|\beta_i^{\text{feat}} - \beta_j^{\text{feat}}\|_2 \leq \mathscr{L}\|e_i - e_j\|_2$ for two graph $G_i$ and $G_j$, so nearby domains yield similar feature transforms and thus neighboring subspaces in the unified feature space.*

The domain-conditioned transformations $\{\mathcal{K}_i^{\text{feat}}\}_{i=1}^M$ parameterized by shared $\phi_{\text{feat}}$, are learned jointly over all pre-training graphs $\mathcal{G}$ using their domain embeddings $\{e_i\}_{i=1}^M$ to form a unified and general feature space. Together with the GNN encoder $f_\theta$, they constitute part of the pretrained model $\mathcal{M}$ (i.e., $\theta, \phi_{\text{feat}} \in \Phi$).

**In-context Feature Alignment.** At test time, for a downstream graph $G_{\text{new}}$ with domain embedding $e_{\text{new}}$ computed via the pretrained domain embedder, we generate its alignment parameters $(\gamma_{\text{new}}^{\text{feat}}, \beta_{\text{new}}^{\text{feat}}) = f_{\phi_{\text{feat}}}(e_{\text{new}})$. For any item $w \in G_{\text{new}}$, its aligned feature is computed as:

$$z_{\text{new},w} = \gamma_{\text{new}}^{\text{feat}} \odot f_\theta(w, G_{\text{new}}) + \beta_{\text{new}}^{\text{feat}}. \tag{11}$$

### 3.2.2 LABEL ALIGNMENT

Different graphs have their own label systems, leading to label IDs across domains having inconsistent semantics, such as label ID 0 representing different concepts across graphs. To reconcile graph-local label systems, we maintain a shared label base $\mathbf{E}^{\text{label}} \in \mathbb{R}^{L_{\max} \times d}$ with

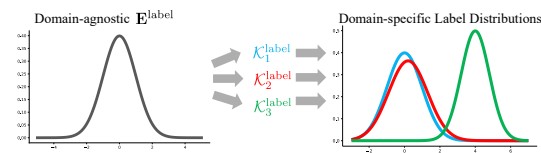

Figure 3: Domain-conditioned label alignment.

$L_{\max} = \max_i C_i$, where each row $\mathbf{E}_l^{\text{label}}$ serves as a domain-agnostic label prototype for ID $l$, initialized as $\mathbf{E}_l^{\text{label}} \sim \mathcal{N}(0, \mathbf{I}_d)$. Given domain embedding $e_i$ of $G_i$, we instantiate a domain-conditioned label transformation with FiLM, architecturally identical to the feature-side transformation, that maps $e_i$ to scale and shift parameters for label alignment:

$$
\begin{aligned}
\left(\gamma_i^{\text{label}}, \beta_i^{\text{label}}\right) &= f_{\phi_{\text{label}}}(e_i), \quad \gamma_i^{\text{label}}, \beta_i^{\text{label}} \in \mathbb{R}^d, \\
u_{i,l} &= \mathcal{K}_i^{\text{label}}(\mathbf{E}_l) = \gamma_i^{\text{label}} \odot \mathbf{E}_l^{\text{label}} + \beta_i^{\text{label}}, \quad l \in L_i = \{0, \cdots, C_i - 1\},
\end{aligned}
\tag{12}
$$

where $f_{\phi_{\text{label}}} : \mathbb{R}^d \to \mathbb{R}^{2d}$ is a two-layer MLP with a SoftPlus for $\gamma_i^{\text{label}}$. $u_{i,l}$ is the aligned label embeddings conditioned on $e_i$ for label $l$. As illustrated in Fig. 3, this mechanism transforms a single domain-agnostic distribution into domain-specific label distributions. The shared base $\mathbf{E}^{\text{label}}$ provides a common reference, while FiLM parameters shift and scale these prototypes based on domain characteristics, ensuring semantically distinct labels occupy different subspaces in a unified label space even when sharing the same ID. $f_{\phi_{\text{label}}}$ is a component of the pretrained model $\mathcal{M}$ ($\phi_{\text{label}} \in \Phi$).

**In-context Label Alignment.** For a new graph $G_{\text{new}}$ with $C_{\text{new}}$ classes, we compute label alignments using the pretrained transformation $(\gamma_{\text{new}}^{\text{label}}, \beta_{\text{new}}^{\text{label}}) = f_{\phi_{\text{label}}}(e_{\text{new}})$. The aligned label embeddings for $G_{\text{new}}$ are:

$$\mathbf{u}_{\text{new},l} = \gamma_{\text{new}}^{\text{label}} \odot \mathbf{E}_l^{\text{label}} + \beta_{\text{new}}^{\text{label}}, \quad l \in \{0, \ldots, C_{\text{new}} - 1\} \tag{13}$$

which yields domain-aware label prototypes that are compatible with the unified feature space and ready for few-shot matching.

### 3.3 EPISODIC PRETRAINING

MF-GIA is pretrained with an episodic, prompt-based objective that teaches the model to match aligned item features to aligned label prototypes, mimicking the few-shot scenarios encountered during inference.

For each pretraining graph $G_i \in \mathcal{G}$, we construct $m$-way $k$-shot episodes to simulate in-context learning scenarios. Specifically, in each episode, we select $m$ classes and sample $k$ labeled items per class to form a support set $\mathcal{S} = \bigcup_{c=1}^{m} \left\{ \left( w_j^{(c)}, l^{(c)} \right) \right\}_{j=1}^{k}$, where $w_j^{(c)}$ is the $j$-th item of the $c$-th selected class and $l^{(c)} = y_j$ is its label ID. We also sample $T$ items per class for the query set $\mathcal{Q} = \bigcup_{c=1}^{m} \left\{ q_t^{(c)}, l^{(c)} \right\}_{t=1}^{T}$. Using the domain embedding $e_i$, we compute aligned item features with Eq. (10) and aligned label prototypes with Eq. (12), yielding $z_{i,w_j^{(c)}} = \mathcal{K}_i^{\text{feat}}\left( h_{i,w_j^{(c)}} \right)$, $z_{i,q_t^{(c)}} = \mathcal{K}_i^{\text{feat}}\left( h_{i,q_t^{(c)}} \right)$, and $u_{i,l^{(c)}} = \mathcal{K}_i^{\text{label}}\left( \mathbf{E}_{l^{(c)}}^{\text{label}} \right)$. The prompt-query pairs become:

$$\text{Prompt } \mathcal{S} : \left\{ \left( z_{i,w_j^{(c)}}, u_{i,l^{(c)}} \right) \right\}_{c \in [m], j \in [k]}, \quad \text{Query } \mathcal{Q} : \left\{ z_{i,q_t^{(c)}} \right\}_{c \in [m], t \in [T]}. \tag{14}$$

Recalling the episodic meta learning objective in Eq. (2), which requires matching queries to classes using only the prompt, we propose a **Dual Prompt-Aware Attention** (DPAA) mechanism. It allows queries to attend to prompt examples but prevents prompts from interacting with each other, strictly following the principle of in-context learning. Specifically, let $\mathbf{Z}^{\text{pmt}} = \left[ z_{i,w_1^{(1)}}, \cdots, z_{i,w_k^{(m)}} \right] \in \mathbb{R}^{mk \times d}$ be the matrix of row-stacked support features and $\mathbf{U}^{\text{pmt}} = \left[ u_{i,l^{(1)}}, \cdots, u_{i,l^{(m)}} \right] \in \mathbb{R}^{m \times d}$ be the label prototype matrix. DPAA consists of two single-query attention layers, one feature-side and one label-side, both sharing the same projection matrices $\mathbf{W}_K, \mathbf{W}_V \in \mathbb{R}^{d \times d}$. For an aligned query feature $z_{i,q} \in \mathcal{Q}$ from $G_i$, the feature-side attention computes:

$$\mathbf{K}^{\text{feat}} = \mathbf{Z}^{\text{pmt}} \mathbf{W}_K, \quad \mathbf{V}^{\text{feat}} = \mathbf{Z}^{\text{pmt}} \mathbf{W}_V, \quad \mathbf{Q}^{\text{feat}} = z_{i,q} \mathbf{W}_Q,$$
$$z_{i,q}^{\text{out}} = \text{softmax}\left( \frac{\mathbf{Q}^{\text{feat}}(\mathbf{K}^{\text{feat}})^\top}{\sqrt{d}} \right) \mathbf{V}^{\text{feat}}, \tag{15}$$

where the attended representation $z_{i,q}^{\text{out}}$ aggregates features from prompt examples relevant to the query. In other words, Eq. (15) aims to use the support features $\mathbf{Z}^{\text{pmt}}$ to prompt the query feature $z_{i,q}$, producing the prompt-conditioned feature $z_{i,q}^{\text{out}}$ for the query. $z_{i,q}^{\text{out}}$ is then projected to label space via a learnable function $f_\Omega : \mathbb{R}^d \to \mathbb{R}^d$, which is also prompted by the support set. Thus, the label-side attention lets the query interact with label prototypes:

$$\mathbf{K}^{\text{label}} = \mathbf{U}^{\text{pmt}} \mathbf{W}_K, \quad \mathbf{V}^{\text{label}} = \mathbf{U}^{\text{pmt}} \mathbf{W}_V, \quad \mathbf{Q}^{\text{label}} = f_\Omega(z_{i,q}^{\text{out}}),$$
$$u_{i,q}^{\text{out}} = \text{softmax}\left( \frac{\mathbf{Q}^{\text{label}}(\mathbf{K}^{\text{label}})^\top}{\sqrt{d}} \right) \mathbf{V}^{\text{label}}, \tag{16}$$

Analogous to LLMs, where prompt examples guide task completion, here $(\mathbf{Z}^{\text{pmt}}, \mathbf{U}^{\text{pmt}})$ serves as the few-shot demonstrations, $z_{i,q}$ as the query to be answered, and $z_{i,q}^{\text{out}}$ as the prompt-conditioned intermediate, and $u_{i,q}^{\text{out}}$ as the answer produced from the prompt for the task objective $z_{i,q}$. Thus, the pretraining objective is to build the matching between $u_{i,q}^{\text{out}}$ and the ground-truth label. The final prediction is obtained by scoring the query's prompted representation against all label prototypes:

$$s_{i,q} = u_{i,q}^{\text{out}}(\mathbf{U}^{\text{pmt}})^\top \in \mathbb{R}^m, \tag{17}$$

where $s_{i,q}$ contains the per-class scores for the query item $q$. For each episode from $G_i$, we minimize the cross-entropy loss over all queries in $\mathcal{Q}$:

$$\mathcal{L}_{\text{episode}}(G_i) = -\frac{1}{mT} \sum_{c=1}^{m} \sum_{t=1}^{T} \log \frac{\exp\left(s_{i,q_t^{(c)}}[c]/\tau\right)}{\sum_{j=1}^{m} \exp\left(s_{i,q_t^{(c)}}[j]/\tau\right)}, \tag{18}$$

where $\tau > 0$ is a temperature that controls the sharpness of the softmax, $s_{i,q_t^{(c)}}[c]$ denotes the score of the ground-truth class for the query $q_t^{(c)}$. The complete pretraining loss aggregates episodes across all pretraining graphs:

$$\mathcal{L}_{\text{pretrain}} = \mathbb{E}_{G_i \sim \mathcal{G}} \mathbb{E}_{\text{episode} \sim G_i} \left[\mathcal{L}_{\text{episode}}(G_i)\right]. \tag{19}$$

Note that the domain embedder $f_{\phi_{\text{de}}}$ is optimized with $\mathcal{L}_{\text{de}}$ prior to episodic pretraining and then kept fixed. Overall, the pretraining model $\mathcal{M}_{\Phi}$ comprises the frozen encoder initialization $f_{\theta_0}$, the domain embedder $f_{\phi_{\text{de}}}$, the domain-conditioned transformation $f_{\phi_{\text{feat}}}$ and $f_{\phi_{\text{label}}}$, the DPAA projection matrices and the projection head $f_{\Omega}$. This episodic regime trains the model to leverage prompt examples for prediction, establishing the feature-label matching capability essential for ICL on unseen domains.

## 3.4 In-context Prediction on Unseen Domains

At test time, MF-GIA freezes all pretrained parameters. Given an unseen graph $G_{\text{new}}$ together with a $C_{\text{new}}$-way $k$-shot support set $\mathcal{S} = \{(w_i, y_i)\}_{i=1}^{k \cdot C_{\text{new}}}$ as prompts, we compute the in-context domain embedding $e_{\text{new}}$ using the gradient fingerprint and the pretrained domain embedder described in Section 3.1. With the pretrained domain-conditioned transformations, any item $w \in G_{\text{new}}$ is mapped to its aligned feature $z_{\text{new},w}$ via Eq. (11). For label IDs $l \in L_{\text{new}} = \{0, \cdots, C_{\text{new}} - 1\}$, the aligned label prototypes are $\{u_{\text{new},l}\}_{l \in L_{\text{new}}}$ obtained by Eq. (13). We then form the prompt matrices $\mathbf{Z}_{\text{new}}^{\text{pmt}} = \left[z_{\text{new},w_1^{(1)}}, \ldots, z_{\text{new},w_k^{(C_{\text{new}})}}\right] \in \mathbb{R}^{(kC_{\text{new}}) \times d}$ and $\mathbf{U}_{\text{new}}^{\text{pmt}} = \left[u_{\text{new},l^{(1)}}, \ldots, u_{\text{new},l^{(C_{\text{new}})}}\right] \in \mathbb{R}^{C_{\text{new}} \times d}$. To make a prediction on a query item $q$, we apply the pretrained DPAA on it. Specifically, the feature-side attention produces the prompt-conditioned feature $z_{\text{new},q}^{\text{out}}$ by Eq. (15), which is then fed to the label-side attention Eq. (16) to yield $u_{\text{new},q}^{\text{out}}$. The final scores and prediction are:

$$
\begin{aligned}
s_{\text{new},q} &= u_{\text{new},q}^{\text{out}} \left(\mathbf{U}_{\text{new}}^{\text{pmt}}\right)^{\top} \in \mathbb{R}^{C_{\text{new}}}, \\
\hat{y}_q &= \arg \max_{j \in [C_{\text{new}}]} s_{\text{new},q}[j].
\end{aligned}
\tag{20}
$$

This inference procedure is parameter-update-free w.r.t. the pretrained model $\mathcal{M}$, and the same pipeline applies to node or edge items by letting $w$ range over $V_{\text{new}}$ or $E_{\text{new}}$. The detailed algorithms and complexity analysis of MF-GIA are provided in Appendix C.

To fully exploit the sparse few-shot support labels and the topology of $G_{\text{new}}$ during inference, we further enhance prototype construction and prediction refinement beyond DPAA. Specifically, we construct graph-aware class prototypes by propagating soft label distributions initialized from the support set through the graph structure using label propagation, which enriches the prototypes with neighborhood context to compensate for the sparsity of the $k$-shot support. For query classification, we employ an adaptive distance metric that combines cosine similarity and inverse Euclidean distance, weighted per query by $\sigma(\text{Var}(z_{\text{new},q}))$, enabling the metric to adapt to local feature geometry. Finally, predictions are iteratively refined through semi-supervised label propagation, where query pseudo-labels are updated by blending propagated and current distributions while keeping support labels fixed, enforcing graph-level consistency in the final output.

## 4 Experiments

### 4.1 Experimental Setup

We employ cross-domain graph datasets to evaluate MF-GIA, which is pretrained exclusively on node classification tasks using four datasets: WikiCS (web link), PubMed and ogbn-Arxiv (citation), and Amazon-ratings (e-commerce rating). These datasets are pre-encoded with heterogeneous feature spaces and label systems, enabling us to learn domain alignment without modality priors.

Table 2: Few-shot node classification accuracy (%) with standard deviation over 10 runs. Best and second-best results are shown in **bold** and underlined. "–" denotes datasets where only encoded features and indexed labels are available, making modality-dependent models inapplicable.

| Method | Cora-7 way *Citation* | | | ogbn-Products-47 way *E-commerce* | | | Computers-10 way *E-commerce* | | | Physics-5 way *Co-authorship* | | | BlogCatalog-6 way *Social Media* | | |
|---|---|---|---|---|---|---|---|---|---|---|---|---|---|---|---|
| | 1-shot | 3-shot | 5-shot | 1-shot | 3-shot | 5-shot | 1-shot | 3-shot | 5-shot | 1-shot | 3-shot | 5-shot | 1-shot | 3-shot | 5-shot |
| GCN | $43.07_{\pm7.37}$ | $42.38_{\pm7.42}$ | $42.55_{\pm2.09}$ | $8.27_{\pm1.48}$ | $7.85_{\pm1.62}$ | $8.77_{\pm1.71}$ | $36.42_{\pm6.28}$ | $39.33_{\pm6.87}$ | $41.09_{\pm6.35}$ | $65.43_{\pm5.12}$ | $73.28_{\pm4.44}$ | $77.15_{\pm3.96}$ | $43.22_{\pm3.95}$ | $49.08_{\pm3.02}$ | $52.16_{\pm2.88}$ |
| GAT | $46.12_{\pm7.10}$ | $47.31_{\pm7.58}$ | $47.71_{\pm8.66}$ | $7.14_{\pm1.55}$ | $7.90_{\pm1.74}$ | $8.39_{\pm1.86}$ | $37.15_{\pm6.43}$ | $40.27_{\pm6.92}$ | $42.03_{\pm6.61}$ | $66.80_{\pm5.05}$ | $75.22_{\pm4.31}$ | $78.41_{\pm3.88}$ | $\underline{46.37_{\pm3.88}}$ | $52.47_{\pm3.10}$ | $56.42_{\pm2.76}$ |
| GraphSAGE | $40.50_{\pm6.11}$ | $42.07_{\pm6.12}$ | $42.40_{\pm6.12}$ | $7.36_{\pm1.68}$ | $8.59_{\pm1.74}$ | $9.42_{\pm1.70}$ | $35.89_{\pm6.34}$ | $38.76_{\pm6.79}$ | $40.58_{\pm6.41}$ | $67.12_{\pm5.25}$ | $71.95_{\pm4.57}$ | $77.36_{\pm4.02}$ | $40.56_{\pm4.02}$ | $53.12_{\pm3.21}$ | $58.03_{\pm2.93}$ |
| GraphMAE | $42.41_{\pm6.38}$ | $43.36_{\pm6.94}$ | $44.22_{\pm6.49}$ | $8.58_{\pm1.63}$ | $9.87_{\pm1.69}$ | $9.94_{\pm1.71}$ | $40.86_{\pm6.31}$ | $42.72_{\pm6.83}$ | $43.35_{\pm6.51}$ | $68.23_{\pm4.89}$ | $77.04_{\pm3.92}$ | $80.35_{\pm3.51}$ | $43.25_{\pm3.71}$ | $57.93_{\pm2.86}$ | $62.14_{\pm2.64}$ |
| DGI | $41.28_{\pm6.54}$ | $42.18_{\pm6.75}$ | $43.27_{\pm6.33}$ | $9.04_{\pm1.49}$ | $10.08_{\pm1.52}$ | $10.87_{\pm1.59}$ | $39.91_{\pm6.22}$ | $41.77_{\pm6.75}$ | $45.54_{\pm6.39}$ | $66.12_{\pm5.01}$ | $74.83_{\pm4.20}$ | $78.09_{\pm3.76}$ | $42.11_{\pm3.79}$ | $56.08_{\pm2.94}$ | $60.91_{\pm2.71}$ |
| GraphCL | $40.22_{\pm6.47}$ | $44.68_{\pm6.88}$ | $45.56_{\pm6.42}$ | $11.93_{\pm1.65}$ | $11.26_{\pm1.71}$ | $13.14_{\pm1.77}$ | $38.74_{\pm6.39}$ | $41.55_{\pm6.91}$ | $43.19_{\pm6.58}$ | $74.35_{\pm4.95}$ | $82.12_{\pm4.05}$ | $\underline{85.40_{\pm3.62}}$ | $44.87_{\pm3.66}$ | $57.20_{\pm2.90}$ | $63.55_{\pm2.69}$ |
| GCOPE | $42.63_{\pm6.33}$ | $43.89_{\pm6.77}$ | $44.74_{\pm6.36}$ | $11.18_{\pm1.60}$ | $11.73_{\pm1.68}$ | $12.54_{\pm1.72}$ | $\underline{43.02_{\pm6.36}}$ | $43.84_{\pm6.87}$ | $47.46_{\pm6.55}$ | $\underline{76.18_{\pm4.81}}$ | $\underline{84.65_{\pm3.88}}$ | $85.07_{\pm3.42}$ | $45.02_{\pm3.58}$ | $\underline{58.76_{\pm2.81}}$ | $63.05_{\pm2.61}$ |
| GPF | $41.12_{\pm6.45}$ | $40.26_{\pm6.84}$ | $43.16_{\pm6.41}$ | $11.12_{\pm1.57}$ | $12.65_{\pm1.63}$ | $13.43_{\pm1.70}$ | $37.02_{\pm6.28}$ | $39.84_{\pm6.79}$ | $41.62_{\pm6.48}$ | $69.28_{\pm4.95}$ | $76.91_{\pm4.22}$ | $83.85_{\pm3.73}$ | $43.08_{\pm3.63}$ | $58.01_{\pm2.83}$ | $63.47_{\pm2.62}$ |
| All in One | $42.66_{\pm6.38}$ | $43.92_{\pm6.81}$ | $44.78_{\pm6.37}$ | $8.15_{\pm1.54}$ | $8.77_{\pm1.61}$ | $8.83_{\pm1.68}$ | $35.64_{\pm6.36}$ | $40.48_{\pm6.82}$ | $44.07_{\pm6.50}$ | $73.43_{\pm5.10}$ | $81.36_{\pm4.37}$ | $85.20_{\pm3.85}$ | $42.54_{\pm3.75}$ | $56.72_{\pm2.96}$ | $61.31_{\pm2.70}$ |
| GFT | $41.40_{\pm6.04}$ | $43.31_{\pm1.11}$ | $43.55_{\pm7.43}$ | $11.12_{\pm1.57}$ | $\underline{14.65_{\pm1.63}}$ | $15.43_{\pm1.70}$ | – | – | – | – | – | – | – | – | – |
| AutoGFM | $\underline{46.29_{\pm7.24}}$ | $47.33_{\pm7.80}$ | $47.76_{\pm8.06}$ | – | – | – | – | – | – | – | – | – | – | – | – |
| Prodigy | $43.27_{\pm6.52}$ | $42.23_{\pm7.65}$ | $44.29_{\pm5.50}$ | $9.53_{\pm1.69}$ | $10.89_{\pm2.01}$ | $11.46_{\pm1.74}$ | $40.29_{\pm6.87}$ | $41.03_{\pm7.52}$ | $45.82_{\pm5.61}$ | $67.26_{\pm7.33}$ | $71.98_{\pm5.25}$ | $79.47_{\pm4.62}$ | $39.85_{\pm3.97}$ | $46.56_{\pm2.62}$ | $53.44_{\pm2.78}$ |
| OFA | $30.38_{\pm2.39}$ | $36.03_{\pm2.11}$ | $32.10_{\pm1.79}$ | $7.42_{\pm1.44}$ | $7.98_{\pm1.51}$ | $8.66_{\pm1.60}$ | – | – | – | – | – | – | – | – | – |
| GraphAlign | $44.37_{\pm8.64}$ | $\underline{48.96_{\pm8.25}}$ | $\underline{52.64_{\pm7.53}}$ | $\underline{12.42_{\pm1.62}}$ | $13.07_{\pm1.69}$ | $\underline{15.92_{\pm1.73}}$ | – | – | – | – | – | – | – | – | – |
| **MF-GIA** | $\mathbf{47.64_{\pm8.77}}$ | $\mathbf{57.38_{\pm9.02}}$ | $\mathbf{63.98_{\pm7.13}}$ | $\mathbf{16.86_{\pm2.55}}$ | $\mathbf{19.16_{\pm2.19}}$ | $\mathbf{22.61_{\pm1.71}}$ | $\mathbf{41.49_{\pm7.49}}$ | $\mathbf{46.21_{\pm14.16}}$ | $\mathbf{53.71_{\pm3.28}}$ | $\mathbf{79.12_{\pm11.54}}$ | $\mathbf{86.48_{\pm0.96}}$ | $\mathbf{88.92_{\pm0.84}}$ | $\mathbf{49.46_{\pm4.02}}$ | $\mathbf{62.69_{\pm2.53}}$ | $\mathbf{67.31_{\pm2.60}}$ |

Table 3: Few-shot edge classification accuracy (%) with standard deviation over 20 episodes.

| Method | FB15K237-5 way *Encyclopedic KG* | | | FB15K237-10 way *Encyclopedic KG* | | | FB15K237-40 way *Encyclopedic KG* | | | WN18RR-5 way *Lexical KG* | | | WN18RR-10 way *Lexical KG* | | |
|---|---|---|---|---|---|---|---|---|---|---|---|---|---|---|---|
| | 1-shot | 3-shot | 5-shot | 1-shot | 3-shot | 5-shot | 1-shot | 3-shot | 5-shot | 1-shot | 3-shot | 5-shot | 1-shot | 3-shot | 5-shot |
| GCN | $82.45_{\pm1.20}$ | $84.30_{\pm1.05}$ | $85.12_{\pm0.98}$ | $70.28_{\pm3.50}$ | $74.61_{\pm2.85}$ | $78.94_{\pm2.43}$ | $55.72_{\pm0.95}$ | $58.11_{\pm0.78}$ | $60.05_{\pm0.70}$ | $32.14_{\pm2.90}$ | $38.62_{\pm2.35}$ | $42.57_{\pm2.01}$ | $24.05_{\pm1.60}$ | $28.93_{\pm1.42}$ | $31.20_{\pm1.35}$ |
| GraphSAGE | $83.11_{\pm1.14}$ | $85.05_{\pm1.01}$ | $86.02_{\pm0.92}$ | $71.36_{\pm3.28}$ | $75.43_{\pm2.71}$ | $79.45_{\pm2.36}$ | $56.40_{\pm0.90}$ | $58.86_{\pm0.76}$ | $60.72_{\pm0.68}$ | $33.05_{\pm2.80}$ | $39.41_{\pm2.26}$ | $43.28_{\pm1.96}$ | $24.83_{\pm1.55}$ | $29.62_{\pm1.39}$ | $31.88_{\pm1.31}$ |
| DGI | $84.22_{\pm1.08}$ | $85.67_{\pm0.96}$ | $86.30_{\pm0.88}$ | $72.48_{\pm3.10}$ | $76.58_{\pm2.59}$ | $80.12_{\pm2.21}$ | $56.95_{\pm0.88}$ | $59.44_{\pm0.72}$ | $61.31_{\pm0.65}$ | $34.57_{\pm2.75}$ | $40.28_{\pm2.20}$ | $44.05_{\pm1.92}$ | $25.67_{\pm1.50}$ | $30.18_{\pm1.34}$ | $32.41_{\pm1.28}$ |
| GraphMAE | $84.90_{\pm1.02}$ | $86.12_{\pm0.90}$ | $\underline{88.05_{\pm0.83}}$ | $73.35_{\pm2.98}$ | $79.02_{\pm2.47}$ | $83.66_{\pm2.14}$ | $\underline{57.43_{\pm0.85}}$ | $59.96_{\pm0.70}$ | $\underline{61.82_{\pm0.63}}$ | $35.42_{\pm2.70}$ | $41.16_{\pm2.13}$ | $44.83_{\pm1.86}$ | $26.31_{\pm1.47}$ | $30.71_{\pm1.32}$ | $32.95_{\pm1.25}$ |
| GCOPE | $80.12_{\pm1.08}$ | $81.56_{\pm0.96}$ | $82.41_{\pm0.90}$ | $68.03_{\pm2.85}$ | $71.22_{\pm2.43}$ | $74.18_{\pm2.05}$ | $51.08_{\pm0.97}$ | $53.12_{\pm0.85}$ | $56.74_{\pm0.79}$ | $30.47_{\pm3.10}$ | $36.05_{\pm2.62}$ | $40.21_{\pm2.34}$ | $22.18_{\pm1.72}$ | $26.73_{\pm1.55}$ | $29.31_{\pm1.42}$ |
| Prodigy | $87.59_{\pm0.84}$ | $\underline{88.02_{\pm0.48}}$ | $88.05_{\pm0.68}$ | $66.10_{\pm9.89}$ | $\underline{79.61_{\pm8.28}}$ | $\underline{84.30_{\pm7.80}}$ | $54.30_{\pm0.69}$ | $59.58_{\pm0.22}$ | $\mathbf{62.03_{\pm0.59}}$ | $46.57_{\pm6.63}$ | $47.28_{\pm4.06}$ | $53.94_{\pm4.88}$ | $27.01_{\pm2.58}$ | $28.46_{\pm3.77}$ | $33.54_{\pm4.29}$ |
| GFT | $\underline{87.67_{\pm0.89}}$ | $86.00_{\pm1.84}$ | $86.27_{\pm1.10}$ | $\underline{79.17_{\pm1.76}}$ | $79.13_{\pm1.57}$ | $78.83_{\pm1.80}$ | $\mathbf{60.79_{\pm1.41}}$ | $\underline{61.48_{\pm1.32}}$ | $61.12_{\pm1.64}$ | $48.13_{\pm1.87}$ | $48.53_{\pm3.68}$ | $48.80_{\pm3.61}$ | $35.33_{\pm4.20}$ | $\underline{35.50_{\pm5.02}}$ | $\underline{35.50_{\pm4.59}}$ |
| AutoGFM | – | – | – | – | – | – | – | – | – | $\underline{48.47_{\pm4.38}}$ | $\underline{49.10_{\pm3.31}}$ | $49.93_{\pm3.63}$ | $\mathbf{39.34_{\pm3.83}}$ | $\mathbf{39.55_{\pm2.46}}$ | $\mathbf{40.02_{\pm2.26}}$ |
| GraphAlign | $83.02_{\pm1.28}$ | $83.15_{\pm1.07}$ | $84.92_{\pm0.98}$ | $73.25_{\pm3.05}$ | $76.14_{\pm2.58}$ | $77.02_{\pm2.20}$ | $53.10_{\pm8.32}$ | $54.26_{\pm3.93}$ | $59.35_{\pm6.70}$ | $45.08_{\pm10.55}$ | $47.47_{\pm9.88}$ | $\underline{60.19_{\pm10.31}}$ | $27.80_{\pm3.05}$ | $30.65_{\pm2.82}$ | $32.10_{\pm2.60}$ |
| **MF-GIA** | $\mathbf{98.77_{\pm1.03}}$ | $\mathbf{99.42_{\pm0.38}}$ | $\mathbf{99.64_{\pm0.20}}$ | $\mathbf{87.57_{\pm4.03}}$ | $\mathbf{91.24_{\pm2.17}}$ | $\mathbf{91.38_{\pm1.01}}$ | $57.17_{\pm3.79}$ | $\mathbf{61.18_{\pm3.66}}$ | $\mathbf{62.03_{\pm3.79}}$ | $\mathbf{55.64_{\pm10.30}}$ | $\mathbf{64.54_{\pm6.42}}$ | $\mathbf{68.05_{\pm4.39}}$ | $\underline{28.87_{\pm3.89}}$ | $33.12_{\pm4.67}$ | $35.12_{\pm3.80}$ |

For downstream evaluation, we test on both node-level and edge-level classification tasks across seen and unseen domains. For node classification, we evaluate on Cora (citation), and unseen domains including ogbn-Products and Computers (e-commerce product), Physics (co-authorship), and BlogCatalog (social media). For link classification, we assess performance on two knowledge graphs (KGs) from different domains to predict the relation types: FB15K237 (encyclopedic) and WN18RR (lexical), which represent entirely new tasks not encountered in pretraining. To unify the task formulation, we transform the edge-level task into the node-level task by converting edges to nodes in a line graph, as detailed in Appendix D. This task allows us to evaluate our model's generalization capability on an entirely new task and domains not seen during pretraining. More information about baseline configurations, datasets, and implementation details is provided in Appendix E.

## 4.2 IN-CONTEXT LEARNING RESULTS

Table 2 demonstrates that MF-GIA achieves state-of-the-art node classification results across diverse graph domains. Remarkably, MF-GIA reaches 63.98% on Cora with 5-shot prompting, which is an 11.34% absolute improvement over the second-best baseline. Across all 15 configurations, MF-GIA consistently outperforms existing methods with an average margin of 4.2%, despite using pure prompt-based inference without any parameter updates. In contrast, methods like GFT and AutoGFM require extensive fine-tuning on target domains yet still achieve inferior results. This superiority reveals a fundamental insight: when equipped with proper cross-domain alignment, ICL beats fine-tuning. The critical role of alignment is also evident when comparing MF-GIA with Prodigy, which is a true ICL model without domain alignment. MF-GIA consistently outperforms Prodigy on unseen domains, demonstrating that domain embeddings capture domain characteristics for successful cross-domain transfer. Recent modality-dependent GFMs fail on graphs without raw text data (marked "–"), while MF-GIA operates universally on any pre-encoded graphs. Moreover, as shown in Table 3, MF-GIA excels at edge-level tasks, an entirely new task formulation never encountered during pretraining. It demonstrates that MF-GIA captures generalizable patterns for in-context reasoning rather than memorizing dataset/task-specific features. On WN18RR dataset with a 10-way setting, our MF-GIA does not surpass the state-of-the-art baselines GFT and AutoGFM, achieving third-best performance across all shot settings. It is because MF-GIA is pretrained exclusively on node classification tasks, while WN18RR is a dataset for edge-level tasks, which is an entirely different task formulation never encountered during pretraining. We deliberately evaluate on this dataset to assess our model's generalization capacity to unseen tasks, as we believe a genuine graph foundation model should generalize not only to unseen domains but also to unseen task types. While GFT and AutoGFM achieve superior performance on WN18RR-10way, they are pretrained on both node-level and

Table 4: Effect of core components.

| Method | Cora | | Computers | | Physics | |
|---|---|---|---|---|---|---|
| | 1-shot | 5-shot | 1-shot | 5-shot | 1-shot | 5-shot |
| GraphSAGE+FT | 40.50 | 42.40 | 35.89 | 40.58 | 67.12 | 77.36 |
| + Feat. Align. | 42.78 | 45.26 | 37.86 | 45.97 | 75.83 | 85.29 |
| ++ Label Alig. | 43.96 | 49.16 | 37.93 | 47.00 | 76.54 | 87.66 |
| +++ DPAA & $\mathcal{L}_{\text{episode}}$ | **47.64** | **63.98** | **41.49** | **53.71** | **79.12** | **88.92** |

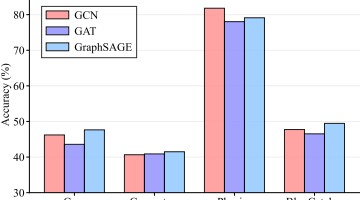

Figure 4: Backbone selections.

edge-level tasks. Therefore, edge classification is not an unseen task for these baselines, so their performance advantage does not necessarily demonstrate stronger cross-task generalization.

### 4.3 MODEL ANALYSIS

**Effect of Core Components.** We analyze the contribution of each component in MF-GIA, starting from its GraphSAGE backbone. GraphSAGE+FT is pretrained on the same datasets and fine-tuned on support sets of test graphs. Adding a domain embedder with FiLM-based feature alignment (+Feat. Align.) improves cross-domain adaptability. Extending alignment to the label space (++Label Align.) further boosts performance by unifying class indices across graphs. Finally, incorporating DPAA with an episodic objective yields the full MF-GIA, which achieves the largest gains across datasets and shots. Table 4 shows a clear step-wise improvement, underscoring that both domain-conditioned alignment and prompt-aware reasoning are crucial for effective graph ICL.

**Effect of Episodic Inference.** ICL can be achieved through two paradigms: episodic meta-learning, which unifies pretraining and inference by training the model to perform inference episodes (MF-GIA and Prodigy), and supervised pretraining with test-time prototype construction, where class prototypes are built from support sets and queries are classified by proximity (GraphAlign). As shown in Table 5, episodic inference (MF-GIA) consistently outperforms the supervised variant (MF-GIA$_{\text{sup}}$).

Table 5: Effect of ICL scheme.

| | Computers | Physics |
|---|---|---|
| MF-GIA$_{\text{sup}}$ | 38.73 | 75.26 |
| MF-GIA | 41.49 | 79.12 |

**Effect of Backbone GNNs.** In MF-GIA, we adopt GraphSAGE as the default backbone. Fig. 4 shows MF-GIA exhibits minor accuracy fluctuations across different GNN backbones under 1-shot settings, demonstrating that MF-GIA is robust to backbone selections. More analytical results are provided in Appendix F.

## 5 CONCLUSION

We introduced MF-GIA, a pretraining framework for graph neural networks that enables in-context learning across heterogeneous domains without relying on modality assumptions. By capturing domain characteristics via gradient fingerprints and aligning pre-encoded features and graph-local labels through domain-conditioned transformations, MF-GIA supports parameter-update-free adaptation from few-shot prompts. This design overcomes key limitations of existing GFMs by removing the need for post-training fine-tuning and modality-specific conversions. Experiments demonstrate strong performance on both seen and unseen domains, with seamless transfer to new tasks.

**Future Directions.** Beyond our current design, MF-GIA opens up several promising avenues for future work. One direction is to couple our gradient fingerprints with LLMs to generate semantic domain descriptions, enabling human-interpretable summaries of latent domain characteristics and more transparent cross-domain reasoning. Another is to leverage these fingerprints to automatically discover latent domain structure from large unlabeled graph collections, moving from manually curated domains to data-driven domain decomposition. We believe these extensions will further enhance the interpretability, automation, and scalability of modality-free graph foundation models.

ACKNOWLEDGEMENTS

This research is supported by the National Research Foundation, Singapore under its Frontier CRP Grant (NRF-F-CRP-2024-0005). Any opinions, findings, and conclusions or recommendations expressed in this material are those of the author(s) and do not reflect the views of the National Research Foundation, Singapore.

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

# A    RELATED WORK

**In-context Learning.**    The modern paradigm of in-context learning (ICL) emerged with GPT-3 (Brown et al., 2020), which demonstrated that large autoregressive transformers can adapt to new tasks using only a few labeled demonstrations, without any parameter updates. This breakthrough sparked extensive research along multiple dimensions. From a theoretical perspective, subsequent work has clarified the fundamental mechanisms underlying ICL. Several studies establish connections between ICL and classical meta learning frameworks, drawing parallels to metric-based few-shot methods, including Matching Networks (Vinyals et al., 2016), Prototypical Networks (Snell et al., 2017), and Model-Agnostic Meta Learning (Finn et al., 2017). More recent theoretical analyses reveal that transformers can implement gradient descent algorithms within their forward pass (Von Oswald et al., 2023; Ren & Liu, 2024; Zhang et al., 2024), effectively learning to optimize in-context. Complementary work by (Garg et al., 2022) demonstrates that transformers can learn entire function classes from context, providing an alternative computational perspective on ICL capabilities. On the methodological front, researchers have developed techniques to enhance ICL performance through improved prompt engineering. Min et al. (2022) introduce MetaICL, which explicitly trains models to perform in-context learning. Practical advances focus on demonstration selection and ordering: retrieval-based methods identify optimal examples (Rubin et al., 2022; Luo et al., 2024), while active selection strategies iteratively refine the demonstration set (Zhang et al., 2022; Qin et al., 2024). The sensitivity of ICL to prompt construction has motivated calibration techniques (Zhao et al., 2021) and continuous prompt optimization (Lester et al., 2021), with recent work revealing substantial impacts from demonstration ordering (Guo et al., 2024). However, extending ICL to graph-structured data presents unique challenges due to the heterogeneous nature of graph domains and the complex interplay between topology, features, and labels.

**Graph Foundation Models.**    Graph Foundation Models (GFMs) have evolved from early self-supervised methods like GraphCL (You et al., 2020) and GraphMAE (Hou et al., 2022) toward comprehensive cross-domain and cross-task generalization. This evolution follows three primary research directions. First, **cross-domain unification** is achieved by using text-attributed graphs (TAGs) as a universal modality. Specifically, OFA (Liu et al., 2024a), GOFA (Kong et al., 2025), and UniGraph (He et al., 2025a) convert graphs to textual representations, then adopt LLMs and GNNs to learn semantic and structural information, respectively. UniGLM (Fang et al., 2025) trains unified language models over multiple TAGs, GraphCLIP (Zhu et al., 2025) aligns graph summaries with language via contrastive learning, and (Wang et al., 2024b) introduces transferable tree vocabularies. AutoGFM  (Chen et al., 2025) studies the automatically adapting architectures to different TAGs. In contrast, text-free methods avoid modality conversion. For example, SAMGPT (Yu et al., 2025) employs learnable domain tokens for domain alignment, and GCOPE (Zhao et al., 2024) connects disparate graphs with virtual nodes to enable cross-domain pretraining. Second, for **prompt-based adaptation**, GraphPrompt (Liu et al., 2023) and All in One (Sun et al., 2023) unify pretraining and downstream tasks via learnable prompts; Prodigy (Huang et al., 2023) enables graph ICL with prompt graphs and shows few-zero transfer to unseen graphs; ARC (Liu et al., 2024b) achieves generalist anomaly detection via contextual cues. Knowledge graph models (Wang et al., 2024a) have particularly benefited from this paradigm. For example, ULTRA (Galkin et al., 2024) learns universal relational representations, KG-ICL (Cui et al., 2024) frames reasoning as prompting, and theoretical analysis (Huang et al., 2025) links expressivity to learned relation motifs. Third, for **multimodal extensions**, emerging work extends GFMs beyond single modalities. UniGraph2 (He et al., 2025b) unifies text and visual features with graph structure, while Graph World Model (Feng et al., 2025) integrates graph-structured states for planning and control.  These advances establish a trajectory toward GFMs that function as universal encoders and reasoners across domains, tasks, and modalities.

Our MF-GIA framework is particularly suited to privacy-constrained settings where schemas and modalities differ across organizations but raw features cannot be shared, such as cross-organization fraud detection over transaction graphs from different banks and multi-hospital patient networks with heterogeneous EHR systems. In such scenarios, each party can locally encode its graph, while MF-GIA uses gradient fingerprints and lightweight domain aligners to align these heterogeneous domains without any modality assumptions, such as TAGs. This complements recent trends in GFMs toward realistic cross-domain deployment, including text-based cross-domain models (Chen et al., 2024b) and graph prompt optimization frameworks like HGMP (Jiao et al., 2025), all of which

highlight the growing need for privacy-preserving, modality-agnostic foundation models in practical applications.

# B PROOFS

## B.1 PROOF OF THEOREM 3.1

The theorem shows that the domain embedder $f_{\phi_{de}}$ acts as a distance-preserving map from the domain space to the embedding space. To prove it, we first give a formal definition of the graph domain and graph distance.

**Definition 2** (Graph Domain). *A graph domain $\mathcal{D}_i$ is characterized by a joint distribution $P_i(G, \mathbf{Y})$ over graphs $G = (V, E, \mathbf{X})$ and labels $\mathbf{Y}$, where the feature distribution $P_i(\mathbf{X})$, label distribution $P_i(\mathbf{Y})$, and structure distribution $P_i(E|V)$ jointly define the domain characteristics.*

**Definition 3** (Graph Distance). *For two graphs $G_i = (V_i, E_i, X_i)$ and $G_j = (V_j, E_j, X_j)$ with normalized adjacency matrices $A_i$ and $A_j$, we define the graph distance as $d_G(G_i, G_j) = \|X_i - X_j\|_F + \|A_i - A_j\|_F$, where we assume $|V_i| = |V_j|$ (padding with isolated nodes if necessary for comparison).*

Then we define the domain distance used in Eq. (7) to measure the inherent similarity between domains.

**Definition 4** (Domain Distance). *The distance between two domains $\mathcal{D}_i$ and $\mathcal{D}_j$ is measured by the Wasserstein distance:*

$$\mathcal{W}_2\left(\mathcal{D}_i, \mathcal{D}_j\right) = \inf_{\kappa \in \Gamma(P_i, P_j)} \left(\mathbb{E}_{(G_i, G_j) \sim \kappa}\left[d_{\mathcal{G}}^2(G_i, G_j)\right]\right)^{1/2}, \tag{21}$$

*where $\Gamma(P_i, P_j)$ denotes all joint distributions with marginals $P_i$ and $P_j$, $d_{\mathcal{G}}(G_i, G_j)$ is a graph distance metric that captures both feature and structural differences between graphs.*

Before proving this theorem, we establish a technical lemma that characterizes the properties of gradient fingerprints.

**Lemma B.1.** *For a graph $G = (V, E, \mathbf{X}, \mathbf{Y})$ and the one-layer GNN encoder initialization $\theta_0 \in \mathbb{R}^{d_o \times d}$, the gradient of the task loss at $\theta_0$ can be decomposed as:*

$$\nabla_\theta \mathcal{L}(\theta_0, G) = \frac{1}{|V|} \mathbf{X}^\top \mathbf{A} \cdot \text{diag}(\mathbf{g}) \cdot \mathbf{1}_d, \tag{22}$$

*where $\mathbf{A}$ is the normalized adjacency matrix, $\mathbf{g} = \left[g_1, \cdots, g_{|V|}\right]^\top \in \mathbb{R}^{|V|}$ is a vector of per-node loss gradients with $g_v = \nabla_{h_v} \ell(h_v, y_v)$, $\ell$ is the node-level loss function, and $\mathbf{1}_d$ is the all-ones vector.*

*Proof.* Consider the forward pass of a one-layer GNN with weight matrix $\theta \in \mathbb{R}^{d_o \times d}$:

$$\mathbf{H} = \sigma(\mathbf{A}\mathbf{X}\theta), \tag{23}$$

where $\sigma$ is the nonlinear activation function (e.g., ReLU). The task loss over the graph is:

$$\mathcal{L}(\theta, G) = \frac{1}{|V|} \sum_{v \in V} \ell(h_v, y_v), \tag{24}$$

where $h_v$ is the representation of $v$, which is the $v$-th row of $\mathbf{H}$, and $\ell$ is the node-level loss function (e.g., cross-entropy). Computing the gradient with respect to $\theta$ via the chain rule:

$$\nabla_\theta \mathcal{L}(\theta, G) = \frac{1}{|V|} \sum_{v \in V} \nabla_\theta h_v^\top \cdot \nabla_{h_v} \ell(h_v, y_v). \tag{25}$$

For the gradient of $h_v$ w.r.t. $\theta$, it can be represented as:

$$\nabla_\theta h_v = \left(\sum_{u \in \mathcal{N}(v)} A_{vu} x_u\right)^\top \otimes \sigma'((\mathbf{A}\mathbf{X}\theta)_v). \tag{26}$$

At initialization $\theta_0$, assuming ReLU activation with appropriate initialization ensuring positive pre-activations, we have $\sigma'\left((\mathbf{A}\mathbf{X}\theta_0)_v\right) \approx \mathbf{1}_d$. Therefore, the gradient can be represented as:

$$\nabla_\theta \mathcal{L}(\theta_0, G) = \frac{1}{|V|} \sum_{v \in V} \left( \sum_{u \in \mathcal{N}(v)} A_{vu} x_u \right)^\top \cdot g_v, \tag{27}$$

where $g_v = \nabla_{h_v} \ell(h_v, y_v)$ is the gradient of the loss with respect to node $v$'s representation. This can be rewritten in matrix form as:

$$\nabla_\theta \mathcal{L}(\theta_0, G) = \frac{1}{|V|} \mathbf{X}^\top \mathbf{A}^\top \cdot \operatorname{diag}(\mathbf{g}) \cdot \mathbf{1}_d. \tag{28}$$

For undirected graphs with symmetric normalized adjacency matrices, $\mathbf{A}^\top = \mathbf{A}$, yielding:

$$\nabla_\theta \mathcal{L}(\theta_0, G) = \frac{1}{|V|} \mathbf{X}^\top \mathbf{A} \cdot \operatorname{diag}(\mathbf{g}) \cdot \mathbf{1}_d, \tag{29}$$

which completes the proof. $\qquad\square$

Building on the above definitions and lemmas, we now present the proof of Theorem 3.1.

*Proof.* Since the gradient fingerprint for $G_i$ sampled from domain $\mathcal{D}_i$, gradient fingerprint is defined as:

$$\Delta\theta_i = \theta_0 - \eta \nabla_\theta \mathcal{L}_i(\theta_0, G_i). \tag{30}$$

For two graphs $G_i$ and $G_j$ from two domains, we have:

$$\Delta\theta_i - \Delta\theta_j = -\eta\left(\nabla_\theta \mathcal{L}_i(\theta_0, G_i) - \nabla_\theta \mathcal{L}_j(\theta_0, G_j)\right). \tag{31}$$

Then the Frobenius norm can be represented as:

$$\|\Delta\theta_i - \Delta\theta_j\|_F = \eta \|\nabla_\theta \mathcal{L}_i(\theta_0, G_i) - \nabla_\theta \mathcal{L}_j(\theta_0, G_j)\|_F. \tag{32}$$

Based on the Lemma B.1, and assuming $|V_i| = |V_j| = n$ (we can pad with isolated nodes if necessary) for simplicity, the gradient difference becomes:

$$\nabla_\theta \mathcal{L}_i(\theta_0, G_i) - \nabla_\theta \mathcal{L}_j(\theta_0, G_j) = \frac{1}{n}\left[\mathbf{X}_i^\top \mathbf{A}_i \cdot \operatorname{diag}(\mathbf{g}_i) - \mathbf{X}_j^\top \mathbf{A}_j \cdot \operatorname{diag}(\mathbf{g}_j)\right] \cdot \mathbf{1}_d, \tag{33}$$

where $\mathbf{g}_i = [g_{i,1}, g_{i,2}, \cdots, g_{i,n}]^\top \in \mathbb{R}^n$ is the vector of per-node loss gradient with $g_{i,v} = \nabla_{h_{i,v}} \ell(h_{i,v}, y_{i,v})$ for node $v$ in $G_i$. Let $\mathbf{M}_i = \mathbf{X}_i^\top \mathbf{A}_i \cdot \operatorname{diag}(\mathbf{g}_i) \in \mathbb{R}^{d_o \times n}$ and $\mathbf{M}_j = \mathbf{X}_j^\top \mathbf{A}_j \cdot \operatorname{diag}(\mathbf{g}_j) \in \mathbb{R}^{d_o \times n}$, taking into Eq. (33) and computing the Frobenius norm, we have:

$$\|\nabla_\theta \mathcal{L}_i(\theta_0, G_i) - \nabla_\theta \mathcal{L}_j(\theta_0, G_j)\|_F = \frac{1}{n} \|(\mathbf{M}_i - \mathbf{M}_j) \cdot \mathbf{1}_d\|_2. \tag{34}$$

Since $\|\mathbf{1}_d\|_2 = \sqrt{d}$, with the property of operator norm, the following inequality holds:

$$\|(\mathbf{M}_i - \mathbf{M}_j) \cdot \mathbf{1}_d\|_2 \le \|\mathbf{M}_i - \mathbf{M}_j\|_{\mathrm{op}} \cdot \|\mathbf{1}_d\|_2 = \|\mathbf{M}_i - \mathbf{M}_j\|_{\mathrm{op}} \cdot \sqrt{d}, \tag{35}$$

where $\|\cdot\|_{\mathrm{op}}$ denotes the operator norm. To bound $\|\mathbf{M}_i - \mathbf{M}_j\|_{\mathrm{op}}$, we decompose it by adding and subtracting intermediate terms as:

$$\begin{aligned} \mathbf{M}_i - \mathbf{M}_j &= \mathbf{X}_i^\top \mathbf{A}_i \cdot \operatorname{diag}(\mathbf{g}_i) - \mathbf{X}_j^\top \mathbf{A}_j \cdot \operatorname{diag}(\mathbf{g}_j) \\ &= (\mathbf{X}_i - \mathbf{X}_j)^\top \mathbf{A}_i \cdot \operatorname{diag}(\mathbf{g}_i) + \mathbf{X}_j^\top (\mathbf{A}_i - \mathbf{A}_j) \cdot \operatorname{diag}(\mathbf{g}_i) \\ &\quad + \mathbf{X}_j^\top \mathbf{A}_j \cdot (\operatorname{diag}(\mathbf{g}_i) - \operatorname{diag}(\mathbf{g}_j)). \end{aligned} \tag{36}$$

Taking the operator norm and using the triangle inequality on Eq. (36), we can bound the $\|\mathbf{M}_i - \mathbf{M}_j\|_{\mathrm{op}}$ with:

$$\begin{aligned} \|\mathbf{M}_i - \mathbf{M}_j\|_{\mathrm{op}} \le &\left\|(\mathbf{X}_i - \mathbf{X}_j)^\top\right\|_{\mathrm{op}} \|\mathbf{A}_i\|_{\mathrm{op}} \|\operatorname{diag}(\mathbf{g}_i)\|_{\mathrm{op}} \\ &+ \left\|\mathbf{X}_j^\top\right\|_{\mathrm{op}} \|(\mathbf{A}_i - \mathbf{A}_j)\|_{\mathrm{op}} \|\operatorname{diag}(\mathbf{g}_i)\|_{\mathrm{op}} \\ &+ \left\|\mathbf{X}_j^\top\right\|_{\mathrm{op}} \|\mathbf{A}_j\|_{\mathrm{op}} \|\operatorname{diag}(\mathbf{g}_i) - \operatorname{diag}(\mathbf{g}_j)\|_{\mathrm{op}}. \end{aligned} \tag{37}$$

For a diagonal matrix, the operator norm is the maximum absolute value of its diagonal entries. Therefore, for the diagonal matrix of the per-node gradient vector $\text{diag}(\mathbf{g}_i)$ of graph $G_i$, its operator norm can be computed by $\|\text{diag}(\mathbf{g}_i)\|_{\text{op}} = \max_{v \in V_i} |g_{i,v}|$. Based on the assumption in Theorem 3.1 that the task loss $\mathcal{L}$ is $\mathscr{L}_{\text{task}}$-smooth with respect to model parameters, it means that its gradient is $\mathscr{L}_{\text{task}}$-Lipschitz continuous. Let the task loss $\mathcal{L}$ be a cross-entropy loss with $C_i$ classes, the node-level loss on node $v$ in $G_i$ is $\ell(h_{i,v}, y_v) = -\log\left(\frac{\exp\left(h_{i,v}^{(y_v)}\right)}{\sum_{c=1}^{C_i} \exp\left(h_{i,v}^{(c)}\right)}\right)$. The gradient on $v$ is $g_{i,v} = \nabla_{h_{i,v}} \ell(h_{i,v}, y_v) = p_v - r_{y_v}$, where $p_v = \text{softmax}(h_{i,v}) \in [0,1]^{C_i}$ and $r_{y_v}$ is the one-hot encoding of the ground-truth label of $v$. Therefore, the following inequality holds:

$$\|g_{i,v}\|_2 = \|p_v - r_{y_v}\|_2 \leq \|p_v\|_2 + \|r_{y_v}\|_2 \leq \sqrt{C_i} + 1. \tag{38}$$

Since we are considering $g_{i,v}$ as a scalar after projection, we have:

$$|g_{i,v}| \leq \sqrt{C_i} + 1 =: \hat{C}_i \tag{39}$$

Thus, we have $\|\text{diag}(\mathbf{g}_i)\|_{\text{op}} \leq \hat{C}_i$. Based on the nature of the operator norm, we have $\left\|(\mathbf{X}_i - \mathbf{X}_j)^\top\right\|_{\text{op}} = \|\mathbf{X}_i - \mathbf{X}_j\|_{\text{op}} \leq \|\mathbf{X}_i - \mathbf{X}_j\|_F$, $\|\mathbf{X}_j^\top\|_{\text{op}} = \|\mathbf{X}_j\|_{\text{op}} \leq B$ (bounded by feature norms), and $\|\mathbf{A}_i\|_{\text{op}}, \|\mathbf{A}_j\|_{\text{op}} \leq 1$ where $\mathbf{A}$ is the normalized adjacency matrix, we get:

$$\|\mathbf{M}_i - \mathbf{M}_j\|_{\text{op}} \leq \hat{C}_i \|\mathbf{X}_i - \mathbf{X}_j\|_F + B\hat{C}_i \|\mathbf{A}_i - \mathbf{A}_j\|_F + B \|\text{diag}(\mathbf{g}_i) - \text{diag}(\mathbf{g}_j)\|_{\text{op}}. \tag{40}$$

For the last term of Eq. (40), since the task loss $\mathcal{L}$ is $\mathscr{L}_{\text{task}}$-smooth, the node-level loss $\ell$ inherits smoothness with Lipschitz gradient constant $\mathscr{L}_\ell$. Given node $v$ with the same index in $G_i$ and $G_j$, the gradient difference between $\mathbf{g}_i$ and $\mathbf{g}_j$ is $\|\text{diag}(\mathbf{g}_i) - \text{diag}(\mathbf{g}_j)\|_{\text{op}} = \max_v |g_{i,v} - g_{j,v}|$. Since $g_{i,v} = \nabla_{h_{i,v}} \ell(h_{i,v}, y_{i,v})$ and the gradient is Lipschitz, we have:

$$\|g_{i,v} - g_{j,v}\| \leq \mathscr{L}_\ell \|h_{i,v} - h_{j,v}\|. \tag{41}$$

In the one-layer message passing defined in Eq. (23), the activation function $\sigma(\cdot)$ is typically ReLU, which is 1-Lipschitz. Thus, we have:

$$\|h_{i,v} - h_{j,v}\| \leq \left\|[(\mathbf{A}_i\mathbf{X}_i - \mathbf{A}_j\mathbf{X}_j)\theta_0]_v\right\| \leq \left\|(\mathbf{A}_i\mathbf{X}_i - \mathbf{A}_j\mathbf{X}_j)_v\right\| \cdot \|\theta_0\|, \tag{42}$$

where the node $v$ in $\mathbf{A}_i\mathbf{X}_i - \mathbf{A}_j\mathbf{X}_j$ can be rewritten as $(\mathbf{A}_i\mathbf{X}_i)_v - (\mathbf{A}_j\mathbf{X}_j)_v = \sum_u A_{i,vu} x_{i,u}^T - \sum_u A_{j,vu} x_{j,u}^T$, which can be further bounded by:

$$\begin{aligned}\|(\mathbf{A}_i\mathbf{X}_i - \mathbf{A}_j\mathbf{X}_j)_v\| &\leq \|(\mathbf{A}_i - \mathbf{A}_j)_v \mathbf{X}_i\| + \|(\mathbf{A}_j)_v(\mathbf{X}_i - \mathbf{X}_j)\| \\ &\leq \|(\mathbf{A}_i - \mathbf{A}_j)_v\|\|\mathbf{X}_i\|_{\text{op}} + \|(\mathbf{A}_j)_v\|\|\mathbf{X}_i - \mathbf{X}_j\|_{\text{op}}.\end{aligned} \tag{43}$$

Since $\mathbf{A}$ is normalized where $\|(\mathbf{A}_j)_v\|_1 = 1$ and thus $\|(\mathbf{A}_j)_v\|_2 \leq 1$. Taking Eq. (43) into Eq. (42), we can bound the difference between $h_{i,v}$ and $h_{j,v}$ as:

$$\|h_{i,v} - h_{j,v}\| \leq \|\theta_0\| \cdot \left(\|(\mathbf{A}_i - \mathbf{A}_j)_v\| B + \|\mathbf{X}_i - \mathbf{X}_j\|_{\text{op}}\right). \tag{44}$$

To get a uniform bound over all nodes, we have:

$$\max_v \|h_{i,v} - h_{j,v}\| \leq \|\theta_0\| \cdot \left(\|\mathbf{A}_i - \mathbf{A}_j\|_F B + \|\mathbf{X}_i - \mathbf{X}_j\|_F\right), \tag{45}$$

which gives us:

$$\|\text{diag}(\mathbf{g}_i) - \text{diag}(\mathbf{g}_j)\|_{\text{op}} \leq \mathscr{L}_\ell \|\theta_0\| \cdot \left(B \|\mathbf{A}_i - \mathbf{A}_j\|_F + \|\mathbf{X}_i - \mathbf{X}_j\|_F\right). \tag{46}$$

Since $\mathscr{L}_\ell$, $\theta_0$, and $B$ are fixed, we can use $\mathscr{L}' = \mathscr{L}_\ell \|\theta_0\| \max(B,1)$ to combine them. There, Eq. (46) can be rewritten as:

$$\|\text{diag}(\mathbf{g}_i) - \text{diag}(\mathbf{g}_j)\|_{\text{op}} \leq \mathscr{L}' \left(\|\mathbf{X}_i - \mathbf{X}_j\|_F + \|\mathbf{A}_i - \mathbf{A}_j\|_F\right). \tag{47}$$

To measure the inherently distance between two graphs, we define the graph distance as $d_{\mathcal{G}}(G_i, G_j) = \|\mathbf{X}_i - \mathbf{X}_j\|_F + \|\mathbf{A}_i - \mathbf{A}_j\|_F$ as Definition 3. Then, taking the graph distance and Eq. (47) into Eq. (40), we have:

$$\|\mathbf{M}_i - \mathbf{M}_j\|_{\text{op}} \leq \widetilde{C} \cdot d_{\mathcal{G}}(G_i, G_j), \tag{48}$$

where $\widetilde{C} = \hat{C}_i + B\hat{C}_i + B\mathscr{L}'$. Recall Eq. (32), Eq. (34), and Eq. (35), we can integrate the used constants, such as $\sqrt{d}$ and $\frac{1}{n}$, into $\widetilde{C}$. Then the difference between the gradient fingerprints of two domains is bounded as:

$$\|\Delta\theta_i - \Delta\theta_j\|_F \leq \eta\widetilde{C} \cdot d_{\mathcal{G}}(G_i, G_j). \tag{49}$$

For the graph distance $d_{\mathcal{G}}$, we have:

$$d_{\mathcal{G}}(G_i, G_j) = \|\mathbf{X}_i - \mathbf{X}_j\|_F + \|\mathbf{A}_i - \mathbf{A}_j\|_F \leq \sqrt{2}\sqrt{\|\mathbf{X}_i - \mathbf{X}_j\|_F^2 + \|\mathbf{A}_i - \mathbf{A}_j\|_F^2}. \tag{50}$$

The Wasserstein distance can be rewritten as:

$$\mathcal{W}_2(\mathcal{D}_i, \mathcal{D}_j) = \inf_{\gamma \in \Gamma(P_i, P_j)} \sqrt{\mathbb{E}_{(G_i, G_j) \sim \kappa}\left[d_{\mathcal{G}}^2(G_i, G_j)\right]} \tag{51}$$

For the optimal coupling $\kappa^*$ that achieves the Wasserstein distance, by Jensen's inequality for the concave square root function, we have:

$$\mathbb{E}_{(G_i, G_j) \sim \kappa^*}\left[d_{\mathcal{G}}(G_i, G_j)\right] \leq \sqrt{\mathbb{E}_{(G_i, G_j) \sim \kappa^*}\left[d_{\mathcal{G}}^2(G_i, G_j)\right]} = \mathcal{W}_2(\mathcal{D}_i, \mathcal{D}_j). \tag{52}$$

Combining Eq. (52) and Eq. (49), the bound of $\|\Delta\theta_i - \Delta\theta_j\|_F$ can be expressed as:

$$\|\Delta\theta_i - \Delta\theta_j\|_F \leq \eta\widetilde{C} \cdot \mathbb{E}[d_{\mathcal{G}}(G_i, G_j)] \leq \eta\widetilde{C} \cdot \mathcal{W}_2(\mathcal{D}_i, \mathcal{D}_j). \tag{53}$$

Eq. (53) shows that **the similarity between gradient fingerprints can effectively reflect the domain similarity**. Recall the domain embedder defined in Eq. (5), we can conduct the Lipschitz analysis on it. First, the $\mathrm{Conv2D}$ with kernel weights $\mathbf{W}_{\mathrm{conv}}$ satisfy $\|\mathrm{Conv2D}(x) - \mathrm{Conv2D}(y)\| \leq \|\mathbf{W}_{\mathrm{conv}}\|_{op} \cdot \|x - y\|$, the $\mathrm{Flatten}$ operation preserves norms $\|\mathrm{Flatten}(\mathbf{A})\|_2 = \|\mathbf{A}\|_F$, and the MLP with $R$-layer weight matrices $\mathbf{W}_1 \cdots, \mathbf{W}_R$ satisfy $\|\mathrm{MLP}(x) - \mathrm{MLP}(y)\| \leq \prod_{r=1}^{R}\|\mathbf{W}_r\|_{op} \cdot \|x-y\|$. Thus, the overall Lipschitz constant of the domain embedder $f_{\phi_{\mathrm{de}}}$ is:

$$\mathscr{L}_{\mathrm{de}} = \|\mathbf{W}_{\mathrm{conv}}\|_{op} \cdot \prod_{r=1}^{R}\|\mathbf{W}_r\|_{op}. \tag{54}$$

Thus, the following Lipschitz inequality holds:

$$\|e_i - e_j\|_2 = \|f_{\mathrm{de}}(\Delta\theta_i) - f_{\mathrm{de}}(\Delta\theta_j)\|_2 \leq \mathscr{L}_{\mathrm{de}}\|\Delta\theta_i - \Delta\theta_j\|_F. \tag{55}$$

We can substitute Eq. (53) into Eq. (55) and merge all constants into $\widetilde{C}$, we can get:

$$\|e_i - e_j\|_2 \leq \widetilde{C} \cdot \mathcal{W}_2(\mathcal{D}_i, \mathcal{D}_j) \tag{56}$$

This shows that the domain embedding function preserves domain relationships: similar domains with small $\mathcal{W}_2$ map to nearby embeddings with small $\|e_i - e_j\|_2$.

$\square$

## B.2 Proof of Property 1

*Proof.* By definition, $f_{\phi_{\mathrm{feat}}}$ being $\mathscr{L}$-Lipschitz means that for all $e, e' \in \mathbb{R}^{d_e}$,

$$\|f_{\phi_{\mathrm{feat}}}(e) - f_{\phi_{\mathrm{feat}}}(e')\|_{2,1} \leq \mathscr{L}\|e - e'\|_2. \tag{57}$$

Applying this with $e = e_i$ and $e' = e_j$ for domain embeddings of $G_i$ and $G_j$, we have:

$$f_{\phi_{\mathrm{feat}}}(e_i) - f_{\phi_{\mathrm{feat}}}(e_j) = \left(\gamma^{\mathrm{feat}}(e_i) - \gamma^{\mathrm{feat}}(e_j), \beta^{\mathrm{feat}}(e_i) - \beta^{\mathrm{feat}}(e_j)\right) = (\Delta\gamma, \Delta\beta). \tag{58}$$

Then using the definition of $\|\cdot\|_{2,1}$ gives:

$$\|\Delta\gamma\|_2 + \|\Delta\beta\|_2 = \|f_{\phi_{\mathrm{feat}}}(e_i) - f_{\phi_{\mathrm{feat}}}(e_j)\|_{2,1} \leq \mathscr{L}\|e_i - e_j\|_2, \tag{59}$$

which is exactly the claimed inequality in Property 1. For the graph $G_i$, the domain-conditioned transformation for its domain $i$ on an item $w$ is $\mathcal{K}_i^{\mathrm{feat}} : \mathbb{R}^d \to \mathbb{R}^d$:

$$\mathcal{K}_i^{\mathrm{feat}} = \gamma_i^{\mathrm{feat}} \odot h + \beta_i^{\mathrm{feat}}. \tag{60}$$

Then for any $h \in \mathbb{R}^d$, the following inequality holds:

$$
\begin{aligned}
\left\| \mathcal{K}_i^{\text{feat}}(h) - T_j^{\text{feat}}(h) \right\|_2 &= \left\| \left( \gamma_i^{\text{feat}} - \gamma_j^{\text{feat}} \right) \odot h + \left( \beta_i^{\text{feat}} - \beta_j^{\text{feat}} \right) \right\|_2 \\
&\leq \left\| \left( \gamma_i^{\text{feat}} - \gamma_j^{\text{feat}} \right) \odot h \right\|_2 + \left\| \beta_i^{\text{feat}} - \beta_j^{\text{feat}} \right\|_2 \\
&\leq \|h\|_\infty \left\| \gamma_i^{\text{feat}} - \gamma_j^{\text{feat}} \right\|_2 + \left\| \beta_i^{\text{feat}} - \beta_j^{\text{feat}} \right\|_2 \\
&\leq \|h\|_2 \left\| \gamma_i^{\text{feat}} - \gamma_j^{\text{feat}} \right\|_2 + \left\| \beta_i^{\text{feat}} - \beta_j^{\text{feat}} \right\|_2 ,
\end{aligned}
\tag{61}
$$

Combining Eq. (59) and Eq. (61) yields:

$$
\left\| \mathcal{K}_i^{\text{feat}}(h) - T_j^{\text{feat}}(h) \right\|_2 \leq \max \{ \|h\|_2, 1 \} \, \mathscr{L} \, \|e_i - e_j\|_2 .
\tag{62}
$$

Thus, as domains move closer in the embedding space, their induced feature transforms move closer uniformly on any set of bounded $\|h\|_2$, so the collections $\{\mathcal{K}_i^{\text{feat}}(h_{i,w})\}_w$ and $\{\mathcal{K}_j^{\text{feat}}(h_{j,w})\}_w$ occupy neighboring subspaces in the unified feature space. □

# C  ALGORITHMS AND COMPLEXITY ANALYSIS

## C.1  ALGORITHMS

Algorithm 1 presents the episodic pretraining procedure for MF-GIA. Algorithm 2 details parameter-update-free in-context inference on unseen graphs. At a high level, pretraining teaches the model to (1) extract a gradient-fingerprint domain embedding from a small support set, (2) map that embedding to domain-conditioned feature and label transforms that place heterogeneous graphs in a shared space, and (3) perform prompt-aware matching between queries and aligned supports. At test time, we reuse the same pipeline with frozen parameters, computing the domain embedding and transforms on-the-fly from a few labeled examples only.

## C.2  COMPLEXITY ANALYSIS

Let $G_i = (V_i, E_i, \mathbf{X}i, \mathbf{Y}i)$ be a pretraining graph with $|V_i|$ nodes and $|E_i|$ edges, GNN encoder width $d$, input width $d_o$, and domain embedding width $d_e$. Offline, the domain embedder computes a single gradient fingerprint per graph by one forward–backward through the shared $J$-layer GNN $f_\theta$ from $\theta_0$, which costs $O(J(|E_i| + |V_i|), d)$ per $G_i$. Each fingerprint $\Delta\theta_i \in \mathbb{R}^{d_o \times d}$ is then embedded via $f_{\phi_{\text{de}}}$, giving $O\left(d_e d_o d\right)$ time. After caching $\Delta\theta_i$, training $f_{\phi_{\text{de}}}$ with the pairwise metric-preserving loss adds $O\left(M^2 d_e\right)$ per epoch for $M$ pretraining graphs. In each episode on $G_i$ with $m$-way $k$-shot support and $T$ queries per way, generating FiLM parameters for domain-conditioned transformations $\left(\gamma_i^{\text{feat}}, \beta_i^{\text{feat}}\right) = f_{\phi_{\text{feat}}}(e_i)$ and $\left(\gamma_i^{\text{label}}, \beta_i^{\text{label}}\right) = f_{\phi_{\text{label}}}(e_i)$ costs $O\left(d_e d\right)$, and applying $\mathcal{K}_i^{\text{feat}}$ to $(mk + mT)$ item embeddings and $\mathcal{K}_i^{\text{label}}$ to $m$ label rows of the shared base $\mathbf{E}^{\text{label}} \in \mathbb{R}^{L_{\max} \times d}$ costs $O((mk + mT)d + md)$. The time complexity of DPAA is $O\left((mk + m)d^2 + mT\left(d^2 + mkd + md\right)\right)$, which is typically secondary to the encoder when $m$ and $k$ are few-shot. Typically, we have $M, J, T, k, m \ll |V_i|, |E_i|, d, d_o, d_e$, therefore the overall time complexity can be represented as $O\left(\sum_{i=1}^{M} \left(|E_i| + |V_i|\right) d + d_e d_o d\right)$, which is linear in graph size.

# D  TASK UNIFICATION

With a bit of notation abuse, in knowledge graphs FB15K237 and WN18RR, link classification aims to predict the relation type $r$ for a given triple $(h, r, t)$, where $h$ and $t$ are head and tail entities, respectively. To leverage our node classification framework for this task, we transform the link classification problem into node classification through line graph construction. Given a knowledge graph $G = (V, E, R, \mathbf{X})$ with entities $V$, edges $E$, relation types $R$, and node feature matrix $\mathbf{X}$, we construct a line graph $\text{LG}(G) = (V_{\text{LG}}, E_{\text{LG}}, \mathbf{X}_{\text{LG}})$, where each edge $\varepsilon = (h, r, t) \in E$ becomes a node $v_\varepsilon \in V_{\text{LG}}$. Two nodes $v_{\varepsilon_i}, v_{\varepsilon_j} \in V_{\text{LG}}$ and are connected if the corresponding edges $\varepsilon_i, \varepsilon_j$ share a common entity (head or tail). Formally, the edge set $E_{\text{LG}}$ is defined as:

$$
E_{\text{LG}} = \left\{ \left( v_{\varepsilon_i}, v_{\varepsilon_j} \right) : \varepsilon_i = (h_i, r_i, t_i), \varepsilon_j = (h_j, r_j, t_j), \{h_i, t_i\} \cap \{h_j, t_j\} \neq \emptyset \right\}
\tag{63}
$$

---

**Algorithm 1:** MF-GIA Pretraining

---

**Input** : Pretraining graphs $\mathcal{G} = \{G_i = (V_i, E_i, \mathbf{X}_i, \mathbf{Y}_i)\}_{i=1}^M$; unified item dim $d_o$; embedding dim $d$; GNN encoder $f_\theta$ with stored initialization $\theta_0$ (kept frozen); base label table $\mathbf{E}^{\text{label}} \in \mathbb{R}^{L_{\max} \times d}$; Domain embedder $f_{\phi_{\text{de}}}$; FiLM aligners $f_{\phi_{\text{feat}}}, f_{\phi_{\text{label}}}$; DPAA params $W_K, W_V, W_Q$ and head $f_\Omega$; Episode spec ($m$-way, $k$-shot, $T$ queries), temperature $\tau$, learning rate $\eta$.

**Output :** Pretrained model $\mathcal{M}_\Phi = \{\theta_0, f_{\phi_{\text{de}}}, f_{\phi_{\text{feat}}}, f_{\phi_{\text{label}}}, W_K, W_V, W_Q, f_\Omega\}$.

1 **(Optional) feature unification.** For each $\mathbf{X}_i$, map to $\mathbb{R}^{d_o}$ via SVD.          // unify dims

2 **Stage A: Train domain embedder $f_{\phi_{\text{de}}}$.**                      // gradient fingerprints

3 **for** $i = 1, \ldots, M$ **do**

4     Compute one gradient step on $G_i$ from $\theta_0$: $\theta_i \leftarrow \theta_0 - \eta \nabla_\theta \mathcal{L}_i(\theta_0)$.

5     Store fingerprint $\Delta\theta_i \leftarrow \theta_i - \theta_0$.

6 **end**

7 **repeat**

8     $e_i \leftarrow f_{\phi_{\text{de}}}(\Delta\theta_i)$ for all $i$.

9     $\mathcal{L}_{\text{de}} \leftarrow \sum_{i,j} \left( \|\Delta\theta_i - \Delta\theta_j\|_F - \|e_i - e_j\|_2 \right)^2$.

10    Update $f_{\phi_{\text{de}}}$ by descending $\nabla\mathcal{L}_{\text{de}}$.

11 **until** *converged*

12 Freeze $f_{\phi_{\text{de}}}$ (and keep $f_\theta$ at $\theta_0$).

13 **Stage B: Episodic pretraining with DPAA (encoder init frozen).**

14 **for** *episodes* **do**

15    Sample a graph $G_i$ and an $m$-way $k$-shot support set $\mathcal{S}$ plus $T$ queries per class as the query set $\mathcal{Q}$ .

16    $e_i \leftarrow f_{\phi_{\text{de}}}(\Delta\theta_i)$; $(\gamma_i^{\text{feat}}, \beta_i^{\text{feat}}) \leftarrow f_{\phi_{\text{feat}}}(e_i)$; $(\gamma_i^{\text{label}}, \beta_i^{\text{label}}) \leftarrow f_{\phi_{\text{label}}}(e_i)$.
      // Aligned features / labels

17    For any item $w$, $h_{i,w} \leftarrow f_{\theta_0}(w, G_i)$, $z_{i,w} \leftarrow \gamma_i^{\text{feat}} \odot h_{i,w} + \beta_i^{\text{feat}}$.

18    For each class $l$ used in the episode, $u_{i,l} \leftarrow \gamma_i^{\text{label}} \odot E_l^{\text{label}} + \beta_i^{\text{label}}$.

19    Form $\mathbf{Z}^{\text{pmt}} \in \mathbb{R}^{(mk) \times d}$ from support $\{z_{i,w}\}$ and $\mathbf{U}^{\text{pmt}} \in \mathbb{R}^{m \times d}$ from $\{u_{i,l}\}$.
      // Dual Prompt-Aware Attention (single-query attention)

20    **for** *each query item $q$ with class $c$* **do**

21        $\mathbf{K}^{\text{feat}} \leftarrow \mathbf{Z}^{\text{pmt}} \mathbf{W}_K$, $\mathbf{V}^{\text{feat}} \leftarrow \mathbf{Z}^{\text{pmt}} \mathbf{W}_V$, $\mathbf{Q}^{\text{feat}} \leftarrow z_{i,q} \mathbf{W}_Q$.

22        $z_{i,q}^{\text{out}} \leftarrow \text{softmax}\left( \frac{\mathbf{Q}^{\text{feat}} \mathbf{K}^{\text{feat}\top}}{\sqrt{d}} \right) \mathbf{V}^{\text{feat}}$.

23        $\mathbf{K}^{\text{label}} \leftarrow \mathbf{U}^{\text{pmt}} \mathbf{W}_K$, $\mathbf{V}^{\text{label}} \leftarrow \mathbf{U}^{\text{pmt}} \mathbf{W}_V$, $\mathbf{Q}^{\text{label}} \leftarrow f_\Omega(z_{i,q}^{\text{out}})$.

24        $u_{i,q}^{\text{out}} \leftarrow \text{softmax}\left( \frac{\mathbf{Q}^{\text{label}} \mathbf{K}^{\text{label}\top}}{\sqrt{d}} \right) \mathbf{V}^{\text{label}}$.

25        $s_{i,q} \leftarrow u_{i,q}^{\text{out}}(\mathbf{U}^{\text{pmt}})^\top \in \mathbb{R}^m$; $\mathcal{L}_{\text{episode}} \leftarrow -\log \frac{\exp(s_{i,q}[c]/\tau)}{\sum_{j=1}^m \exp(s_{i,q}[j]/\tau)}$.

26    **end**

27    Update $\{f_{\phi_{\text{feat}}}, f_{\phi_{\text{label}}}, E^{\text{label}}, \mathbf{W}_K, \mathbf{W}_V, \mathbf{W}_Q, f_\Omega\}$ to minimize the mean query loss $\mathcal{L}_{\text{pretrain}}$ of the episode.

28 **end**

---

For each node $v_\varepsilon \in V_{\text{LG}}$ in the line graph corresponding to edge $\varepsilon = (h, r, t)$, we construct features by aggregating the embeddings of the connected entities and relation as $x_{v_e} = [x_h \| x_t]$, where $[\cdot \| \cdot]$ denotes concatenation. The concatenated features are then projected to the unified dimension using PCA to maintain consistency with the node classification framework. After transformation, link classification becomes a node classification problem on the line graph, where each node (representing an edge in the original graph) needs to be classified into one of $R$ relation types.

---

**Algorithm 2:** MF-GIA Test-time In-context Inference (Parameter-Update-Free w.r.t. $\mathcal{M}_\Phi$)

---

**Input** : Frozen pretrained $\Phi$ from Algorithm 1; unseen graph $G_{\text{new}} = (V_{\text{new}}, E_{\text{new}}, \mathbf{X}_{\text{new}})$
with $C_{\text{new}}$ classes;
$C_{\text{new}}$-way $k$-shot support set $\mathcal{S} = \{(w_j, y_j)\}_{j=1}^{kC_{\text{new}}}$; queries $\mathcal{Q} \subseteq G_{\text{new}} \setminus S$.

**Output :** Predictions $\{\hat{y}_q\}_{q \in Q}$.

```
// In-context domain embedding (from support only)
```
1 Compute a one-step fingerprint from $\theta_0$ on $\mathcal{S}$: $\theta_{\text{new}} \leftarrow \theta_0 - \eta \nabla_\theta \mathcal{L}_{\text{new}}(\theta_0; \mathcal{S})$;
$e_{\text{new}} \leftarrow f_{\phi_{\text{de}}}(\theta_{\text{new}} - \theta_0)$.

```
// Domain-conditioned alignment for G_new
```
2 $(\gamma_{\text{new}}^{\text{feat}}, \beta_{\text{new}}^{\text{feat}}) \leftarrow f_{\phi_{\text{feat}}}(e_{\text{new}}); \ (\gamma_{\text{new}}^{\text{label}}, \beta_{\text{new}}^{\text{label}}) \leftarrow f_{\phi_{\text{label}}}(e_{\text{new}})$.
3 For any item $w$: $h_{\text{new},w} \leftarrow f_{\theta_0}(w, G_{\text{new}})$, $z_{\text{new},w} \leftarrow \gamma_{\text{new}}^{\text{feat}} \odot h_{\text{new},w} + \beta_{\text{new}}^{\text{feat}}$.
4 For $l = 0, \ldots, C_{\text{new}} - 1$: $u_{\text{new},l} \leftarrow \gamma_{\text{new}}^{\text{label}} \odot \mathbf{E}_l^{\text{label}} + \beta_{\text{new}}^{\text{label}}$.
5 Form $\mathbf{Z}_{\text{new}}^{\text{pmt}} \in \mathbb{R}^{(kC_{\text{new}}) \times d}$ from $\{z_{\text{new},w}\}_{(w,y) \in S}$ and $\mathbf{U}_{\text{new}}^{\text{pmt}} \in \mathbb{R}^{C_{\text{new}} \times d}$ from $\{u_{\text{new},l}\}$.

```
// Dual Prompt-Aware Attention inference (no parameter update)
```
6 **for** *each query $q \in Q$* **do**
7 $\quad$ $\mathbf{K}^{\text{feat}} \leftarrow \mathbf{Z}_{\text{new}}^{\text{pmt}} \mathbf{W}_K$, $\mathbf{V}^{\text{feat}} \leftarrow \mathbf{Z}_{\text{new}}^{\text{pmt}} \mathbf{W}_V$, $\mathbf{Q}^{\text{feat}} \leftarrow z_{\text{new},q} \mathbf{W}_Q$.
8 $\quad$ $z_{\text{new},q}^{\text{out}} \leftarrow \text{softmax}\left(\frac{\mathbf{Q}^{\text{feat}} \mathbf{K}^{\text{feat}\top}}{\sqrt{d}}\right) \mathbf{V}^{\text{feat}}$.
9 $\quad$ $\mathbf{K}^{\text{label}} \leftarrow \mathbf{U}_{\text{new}}^{\text{pmt}} \mathbf{W}_K$, $\mathbf{V}^{\text{label}} \leftarrow \mathbf{U}_{\text{new}}^{\text{pmt}} \mathbf{W}_V$, $\mathbf{Q}^{\text{label}} \leftarrow f_\Omega(z_{\text{new},q}^{\text{out}})$.
10 $\quad$ $u_{\text{new},q}^{\text{out}} \leftarrow \text{softmax}\left(\frac{\mathbf{Q}^{\text{label}} \mathbf{K}^{\text{label}\top}}{\sqrt{d}}\right) \mathbf{V}^{\text{label}}$.
11 $\quad$ $s_q \leftarrow u_{\text{new},q}^{\text{out}} (\mathbf{U}_{\text{new}}^{\text{pmt}})^\top$; $\quad$ $\hat{y}_q \leftarrow \arg\max_j s_q[j]$.
12 **end**

---

Table 6: Dataset statistics.

| Usage | Dataset | Domain | Task | #Nodes | #Edges | # Classes |
|---|---|---|---|---|---|---|
| **Pretrain** | WikiCS | Web link | Node | 11,701 | 216,123 | 10 |
| | PubMed | Citation | Node | 19,717 | 44,338 | 3 |
| | ogbn-Arxiv | Citation | Node | 169,343 | 1,166,243 | 40 |
| | Amazon-ratings | E-commerce (Ratings) | Node | 24,492 | 93,050 | 5 |
| **Evaluation** | Cora | Citation | Node | 11,701 | 216,123 | 10 |
| | ogbn-Products | E-commerce (Product Category) | Node | 2,449,029 | 61,859,140 | 47 |
| | Computers | E-commerce (Product Category) | Node | 13,752 | 491,722 | 10 |
| | Physics | Co-authorship | Node | 34,493 | 495,924 | 5 |
| | BlogCatalog | Social Media | Node | 5,196 | 343,486 | 6 |
| | FB15K237 | Encyclopedic KG | Link | 14,541 | 310,116 | 237 |
| | WN18RR | Lexical KG | Link | 40,943 | 93,003 | 11 |

## E EXPERIMENTAL DETAILS

### E.1 DATASETS

We pretrain MF-GIA on four source graphs: WikiCS (Mernyei & Cangea, 2020), PubMed (Yang et al., 2016), ogbn-Arxiv (Hu et al., 2020), and Amazon-ratings (Leskovec & Sosič, 2016; Platonov et al., 2023). The pretrained model is evaluated on five held-out graphs on node-level tasks: Cora (Yang et al., 2016), ogbn-Products (Hu et al., 2020), Computers (Shchur et al., 2018), Physics (Shchur et al., 2018), and BlogCatalog (Yang et al., 2023), spanning citation, e-commerce, co-authorship, and social media domains. The pretrained model is also evaluated on edge-level tasks on FB15K237 (Bordes et al., 2013) and WN18RR (Dettmers et al., 2018), which are knowledge graphs from encyclopedic and lexical domains to predict relation types. Dataset statistics are summarized in Table 6. Note that although Amazon-ratings, ogbn-Products, and Computers are E-commerce networks, they form distinct domains: Amazon-ratings is labeled by average user rating per item, whereas ogbn-Products and Computers use product-category labels. We therefore treat them as separate domains.

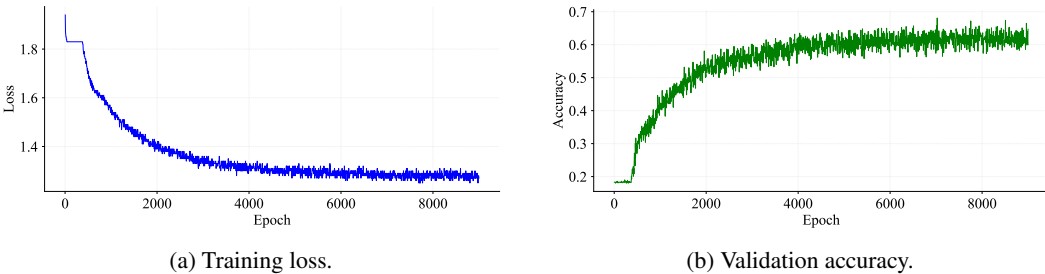

(a) Training loss.          (b) Validation accuracy.

Figure 5: Pretraining curves of MF-GIA.

## E.2 BASELINE CONFIGURATIONS

We compare MF-GIA against two categories of baselines: (1) Traditional GNNs: GCN (Kipf & Welling, 2017), GAT (Veličković et al., 2018), and GraphSAGE (Hamilton et al., 2017); (2) Self-supervised GNNs: GraphMAE (Hou et al., 2022), DGI (Veličković et al., 2019), and GraphCL (You et al., 2020); (3) GFM with post-training: GCOPE (Zhao et al., 2024), GFT (Wang et al., 2024b), AutoGFM (Chen et al., 2025), GPF (Fang et al., 2023), and All in One (Sun et al., 2023); (4) GFM with ICL: Prodigy (Huang et al., 2023), OFA (Liu et al., 2024a), and GraphAlign (Hou et al., 2024).

For traditional GNNs and Self-Supervised Methods, we pretrain on the same four datasets as our MF-GIA. Since these models lack in-context learning capabilities, we fine-tune them on the support set and evaluate on the query set for each episode. For GFMs with post-training (no ICL), we consider two pretraining regimes and report the stronger results: (1) pretraining on the same four graphs as MF-GIA, and (2) pretraining on the datasets used by the methods' official implementations. For Prodigy, we compare two variants: one pretrained on our datasets and another on MAG240M (Hu et al., 2021) as in the original work, reporting the better result. For modality-dependent models (OFA, GraphAlign), which require text-attributed graphs (TAGs) and cannot operate on pre-encoded features, we use their original TAG datasets and implementations. To ensure fairness, all baselines follow the same episode protocol (identical $m$-way, $k$-shot support/query splits), use comparable backbones when applicable, and tune hyperparameters on validation episodes within the authors' recommended ranges.

## E.3 IMPLEMENTATION

We evaluate MF-GIA under the few-shot learning paradigm without any fine-tuning. For each test graph, we randomly sample $k$-shot support sets, where $k = \{1, 3, 5\}$, and evaluate on the remaining nodes. For node classification tasks, we measure classification accuracy and report the mean performance across 10 independent trials, each with randomly sampled support/query splits to ensure robustness of our results. For edge classification tasks, we focus on relation type prediction and conduct evaluation over 20 episodes to account for variance in the few-shot sampling process.

For the dimension alignment, we use SVD to unify all graphs' feature dimensions to $d_o = 64$. The domain embedder uses a 2-layer CNN followed by a 1-layer MLP to project gradient fingerprints into 64-dimensional domain embeddings. For feature alignment, we also employ a 1-layer GNN encoder with a hidden dimension 64, followed by FiLM-based transformations. The label alignment uses a shared label base of dimension $L_{\max} \times 64$. The DPAA mechanism consists of 1 attention layer with 1 head each. During pretraining, we use episodic meta learning with 10-way 5-shot tasks sampled from the pretraining graphs. We train for 10000 episodes using AdamW (Loshchilov & Hutter, 2017) with learning rate 0.005 and weight decay 0.0005. For gradient fingerprint computation, we use a fixed learning rate $\eta = 0.01$ for single-step updates. Fig. 5 illustrates the pretraining dynamics of MF-GIA. The training loss exhibits stable convergence, while validation accuracy shows consistent improvement before plateauing at approximately 6000 epochs, indicating effective model convergence.

Table 7: Effect of pretraining dataset composition on few-shot node classification accuracy (%). We systematically vary domain coverage from single to full four-domain pretraining. Results are 5-shot accuracy averaged over 20 episodes. Best results are **bold**.

| Configuration | Pretraining Datasets | Domain Coverage | In-Domain | Out-of-Domain | | | Avg OOD | Overall |
|---|---|---|---|---|---|---|---|---|
| | | | Cora | Products | Physics | BlogCatalog | | |
| Single | PubMed | Citation | 42.76 | 12.43 | 68.15 | 41.28 | 40.62 | 40.91 |
| | Arxiv | Citation | 48.31 | 13.86 | 76.42 | 43.67 | 44.65 | 45.57 |
| | WikiCS | Web | 39.23 | 14.72 | 65.83 | 45.91 | 42.15 | 41.67 |
| | Amazon | E-commerce | 35.47 | 17.28 | 61.26 | 38.74 | 39.09 | 38.19 |
| Two | PubMed + Arxiv | Citation | 52.14 | 14.92 | 79.87 | 47.35 | 47.38 | 48.57 |
| | WikiCS + Amazon | Web + E-commerce | 41.68 | 18.35 | 67.42 | 48.26 | 44.68 | 43.93 |
| | PubMed + WikiCS | Citation + Web | 47.92 | 15.76 | 73.21 | 49.84 | 46.27 | 46.68 |
| | Arxiv + Amazon | Citation + E-commerce | 50.36 | 19.14 | 78.53 | 46.72 | 48.13 | 48.69 |
| Three | PubMed + Arxiv + WikiCS | w/o E-commerce | 56.47 | 16.82 | 83.74 | 54.38 | 51.65 | 52.85 |
| | PubMed + Arxiv + Amazon | w/o Web | 57.82 | 20.67 | 85.91 | 52.16 | 52.91 | 54.14 |
| | PubMed + WikiCS + Amazon | w/o Citation (Arxiv) | 51.36 | 19.43 | 77.62 | 57.83 | 51.63 | 51.56 |
| | Arxiv + WikiCS + Amazon | w/o Citation (PubMed) | 59.24 | 21.38 | 86.73 | 60.47 | 56.19 | 56.96 |
| Full | All Four Datasets | Complete | **63.98** | **22.61** | **88.92** | **67.31** | **59.61** | **60.73** |

# F MORE EXPERIMENTS

## F.1 EFFECT OF DOMAIN DIVERSITY IN PRETRAINING

Table 7 demonstrates that domain diversity is critical for MF-GIA to construct an effective domain embedding space that enables robust alignment of downstream graphs. Single-dataset pretraining yields limited performance, where Arxiv shows the strongest individual results due to its diverse academic content, providing richer domain signals. When combining two datasets, cross-domain pairs (e.g., Arxiv + Amazon: 48.69%) perform comparably to same-domain pairs (PubMed + Arxiv: 48.57%) despite Amazon's weaker individual performance, indicating that structural diversity helps the domain embedder learn more generalizable alignment patterns beyond surface-level similarities. This effect amplifies with three datasets, where the configuration without PubMed (Arxiv + WikiCS + Amazon) achieves 56.96%, notably outperforming the configuration without Arxiv (51.56%), despite Arxiv's superior standalone performance. This counterintuitive result reveals that once sufficient domain signals are captured, maintaining diverse structural patterns (Web link, E-commerce) becomes more valuable than redundant citation networks for constructing a comprehensive domain space. The full four-dataset model achieves optimal performance (60.73%), with remarkable generalization to entirely unseen domains like BlogCatalog (67.31%), validating that comprehensive domain coverage during pretraining enables the domain embedder to map novel graphs into appropriate subspaces of the learned domain space. These results confirm that MF-GIA's domain-conditioned transformations require diverse pretraining to establish a rich domain embedding space where graphs from any domain, seen or unseen, can be effectively mapped and aligned based on their intrinsic characteristics.

## F.2 EFFECT OF GRAPH PRE-ENCODER

While the raw text data of the Cora dataset has been made available by Chen et al. (2024a), popular end-to-end GNN models typically utilize the pre-encoded version from Yang et al. (2016), which employs bag-of-words (BoW) encoding for node features. Modality-dependent GFMs such as GFT, AutoGFM, and OFA require access to raw text data and rely on specific language models, such as Sentence Transformer (ST) [1] (Reimers & Gurevych, 2019), LLaMa2-7B [2] (Touvron et al., 2023), or RoBERTa [3] (Liu et al., 2019), to align the semantic space between the Cora dataset and their pretraining graphs. In contrast, MF-GIA demonstrates modality freedom, i.e., it operates directly on graphs with arbitrary pre-encoded features without requiring knowledge of the encoding method. Whether a graph has been encoded using bag-of-words, advanced language models,

Table 8: Performance of MF-GIA on Cora with different feature encodings.

| Pre-encoder | 1-shot | 3-shot | 5-shot |
|---|---|---|---|
| BoW | 47.64 | 57.38 | 63.98 |
| ST | 48.37 | 62.79 | 68.54 |
| RoBERTa | 48.24 | 61.53 | 69.85 |
| LLaMa2-7B | 47.93 | 58.64 | 65.33 |

---

[1]https://huggingface.co/sentence-transformers/multi-qa-distilbert-cos-v1
[2]https://huggingface.co/meta-llama/Llama-2-7b
[3]https://huggingface.co/FacebookAI/roberta-base

or even unknown proprietary encoders, MF-GIA can seamlessly adapt to these features. The results in Table 2 employ the public bag-of-words features for MF-GIA. To validate our modality-free claim, in Table 8, we conduct additional experiments applying our pretrained model (without re-pretraining) to the graph pre-encoded through different pipelines. The results show that MF-GIA maintains its effectiveness regardless of the underlying feature encoding, confirming its ability to generalize across diverse pre-processing methods and enabling practical deployment in scenarios where encoding details are unknown or heterogeneous.

### F.3 DIMENSION UNIFICATION

For GFMs, feature dimensions are unified before feeding data into the model because GFMs require inputs in a consistent format. We adopt SVD-based unification to achieve a fair comparison with baselines which isolates the contribution of our core innovation from preprocessing effects. Dedicated methods such as Domain-Invariant Aligner (DIA) (Yuan et al., 2025) provide more expressive and learnable pre-unifiers, which can be seamlessly integrated into our MF-GIA framework. Table 9 shows that our framework remains effective and compatible with advanced dimension unification techniques.

Table 9: Performance on MF-GIA with expressive dimension unification component.

| Method | Computers | | Physics | | BlogCatalog | |
|---|---|---|---|---|---|---|
| | 1-shot | 5-shot | 1-shot | 5-shot | 1-shot | 5-shot |
| MF-GIA | 41.49 | 53.71 | 79.12 | 88.92 | 49.46 | 67.31 |
| MF-GIA (w. DIA) | 41.96 | 54.60 | 80.15 | 89.74 | 48.37 | 66.53 |

### F.4 MORE PRETRAINING TASKS

In this work, we pretrain our model on a small number of datasets to showcase its ability to adapt to diverse unseen domains and task types. The four datasets we use are among the widely adopted, publicly available, and easily accessible graph benchmarks. This design ensures that

Table 10: Performance of MF-GIA with additional pretraining tasks (5-shot).

| | ogbn-Products | Computers | FB15K237 |
|---|---|---|---|
| MF-GIA | 22.61 | 53.71 | 91.38 |
| MF-GIA (w. link) | 24.59 | 56.60 | 92.36 |

anyone can reproduce or extend our pretraining process without the need for extensive dataset curation, as common benchmarks already provide sufficient diversity to pretrain our model effectively. Here, we expand pretraining beyond node classification by adding link existence prediction and re-pretrain the model, denoted as MF-GIA (w. link), and report the inference results under the 5-shot setting in Table 10. It shows that a broader pretraining corpus yields consistent improvements on unseen domains.

### F.5 STABILITY OF GRADIENT FINGERPRINTS

The stability of the gradient fingerprints is dependent on the size of the support set. From Lemma B.1, the gradient decomposes as $\nabla_0 \mathcal{L}(\theta_0, G) = \frac{1}{|V|} \mathbf{X}^\top \mathbf{A} \cdot \mathrm{diag}(\mathbf{g}) \cdot \mathbf{1}_d$, which reveals that the fingerprint aggregates informa-

Table 11: Stability of gradient fingerprints.

| Datasets | Avg. Cosine Similarity | Std. Dev. | Min. Similarity |
|---|---|---|---|
| Physics | 0.931 | 0.015 | 0.896 |
| BlogCatalog | 0.917 | 0.021 | 0.873 |

tion from domain-specific components including feature distribution $\mathbf{X}$, graph structure $\mathbf{A}$ and label distribution $\mathbf{g}$. As the support set size increases, the law of large numbers ensures that these aggregated statistics converge to their population expectations, making the gradient fingerprints increasingly stable. Here, we examine the fingerprint stability under a 5-shot setting. Take Physics and BlogCatalog datasets as examples, we randomly sample 20 different 5-shot support sets from all classes, and compute fingerprints $\Delta\theta$ for each support set and feed it into the domain embedder to get the domain embedding. Then we can measure the stability by computing the pairwise cosine similarity between all domain embedding pairs from the same graph. The results are shown in Table 11, where the high average cosine similarity and low standard deviation demonstrate that different support sets from the same domain produce highly consistent domain embeddings.

Table 12: Effect of $\tau$ on different datasets (5-shot).

| $\tau$ | ogbn-Products | Computers | FB15K237 |
|---|---|---|---|
| 0.05 | 21.45 | 52.10 | 91.82 |
| 0.2 (Default) | 22.61 | 53.71 | 91.38 |
| 0.5 | 22.60 | 53.93 | 90.45 |
| 1 | 22.15 | 53.29 | 90.27 |

Table 13: Sensitivity of DPAA configurations

| | Computers | Physics | BlogCatalog |
|---|---|---|---|
| 1-layer, 1-head (Shared) | 53.71 | 88.92 | 67.31 |
| 1-layer, 1-head (Separated) | 51.25 | 88.10 | 67.05 |
| 1-layer, 4-head | 52.47 | 86.58 | 66.15 |
| 2-layer, 1-head | 54.05 | 88.53 | 67.46 |
| 2-layer, 4-head | 53.39 | 87.52 | 66.83 |

## F.6 SENSITIVITY ANALYSIS

**Sensitivity of the Temperature $\tau$.** The temperature parameter $\tau$ in Eq. (18) controls the sharpness of the softmax distribution over class predictions. We conduct systematic experiments varying $\tau \in \{0.05, 0.2, 0.5, 1\}$ across multiple datasets in Table 12. We can find that the default $\tau = 0.2$ provides optimal balance, but the model is not highly sensitive to this hyperparameter within a reasonable range $[0.2, 1]$. However, as a foundation model, all trainable parameters and hyperparameters must remain fixed across all datasets and tasks. Thus, we set $\tau = 0.2$ uniformly for all experiments both during pretraining and in-context inference without any task- and dataset-specific tuning.

**Sensitivity of DPAA Configurations** For the DPAA depth and number of heads, we directly adopt a single-layer and single-head design for complexity and generalization considerations. Adding more layers and heads would increase computational and memory costs and could introduce a higher risk of overfitting, whereas our goal is to keep the learning process efficient, lightweight, yet expressive. The results Table 2 and Table 3 already show that this simplest DPAA configuration outperforms baselines, which highlights the architectural strength of our approach rather than relying on heavy over-parameterization. Table 13 shows that the 1-layer 1-head and 2-layer 1-head settings achieve the best performance, with 1-layer 1-head offering the best trade-off between efficiency and effectiveness. Therefore, we adopt the 1-layer 1-head configuration as the unified default setting for all datasets in our experiments. Moreover, we conduct an ablation study comparing shared and separate weight matrices $W_K$ and $W_V$ in this table. The shared setting outperforms the separate setting, possibly because the latter introduces more parameters and is more prone to over-fitting, and shared parameters can effectively improve efficiency.

## G THE USE OF LARGE LANGUAGE MODELS (LLMS)

In this paper, LLMs have been used solely for polishing the writing and identifying typographical errors. No LLMs were used for generating research content, analysis, or conclusions.

