# OpenReview forum: "Modality-free Graph In-context Alignment"
_ICLR.cc/2026/Conference — ICLR 2026 Oral_

### Official Review · Reviewer_LCqo · 2025-10-31

**Soundness:** 2
**Presentation:** 2
**Contribution:** 2
**Rating:** 4
**Confidence:** 4

**Summary:**

This paper proposes MF-GIA, a modality-free in-context learning framework for heterogeneous pre-encoded graphs. The key idea is to derive a single-step gradient fingerprint from a frozen GNN initialization and use it as a domain descriptor. This domain embedding modulates lightweight FiLM-based feature and label alignment modules. The method is trained via episodic tasks and performs test-time adaptation by computing a new gradient fingerprint on the support set—without updating model parameters. A dual prompt-aware attention (DPAA) module performs few-shot matching.

**Strengths:**

Novel idea: Leveraging a single-step gradient displacement as a domain signature to instantiate FiLM alignment is an interesting and original approach for graph ICL.
Ambitious scope: The method targets heterogeneous graphs and attempts to unify node-level and edge-level tasks without modality conversion.

**Weaknesses:**

W1. Gradient-fingerprint robustness not validated
The central mechanism assumes that a single-step gradient displacement Δθis a robust domain descriptor. However, the paper provides no sensitivity analysis to different initialization seeds, learning rates, or step counts (1 vs. 2+). If the fingerprint is unstable or dominated by noise, the entire alignment process can break down.

W2. Domain embedding loss collapse risk
The pairwise distance-preserving loss used to train the domain embedder lacks explicit normalization or regularization, and no experiments test whether embedding collapse or trivial scaling happens. While scalability is not a major concern given the small number of pretraining domains, training stability and meaningful domain structure need to be demonstrated.

W3. Misleading “gradient-free” terminology & lack of inference-overhead reporting
The paper repeatedly claims “gradient-free adaptation,” yet inference requires a backward pass to compute Δθ_"new" . This distinction matters for deployment and comparisons against genuinely gradient-free ICL methods. The paper should clarify the terminology (e.g., “parameter-update-free”) and report inference-time overhead (FLOPs, memory, latency).

W4. Label alignment is fragile to label-ID permutation. The FiLM-modulated label prototype table indexes shared label embeddings by label ID. This implicitly assumes consistent label semantics across domains. In practice, label IDs are arbitrary. The paper lacks tests for label-ID permutation invariance or alternative mechanisms to infer label semantics from the support set.

W5. Lack of architectural ablations
No ablation comparing Conv2D vs. more canonical designs (Flatten+MLP, pooling) for Δθembedding. No ablation on DPAA depth or number of heads.

**Questions:**

Please refer to the above weaknesses.

---

> ### Author Response · Authors · 2025-11-19
> **Response to Reviewer LCqo (Part 1/3)**
>
> We sincerely appreciate the time and effort you have invested in reviewing our work and providing valuable feedback. Below we address the issues one by one, and consider all your suggestions in the updated manuscript (highlighted in blue).
>
> > **Q1**: Gradient-fingerprint robustness not validated. The central mechanism assumes that a single-step gradient displacement $\Delta \theta$ is a robust domain descriptor. However, the paper provides no sensitivity analysis to different initialization seeds, learning rates, or step counts (1 vs. 2+). If the fingerprint is unstable or dominated by noise, the entire alignment process can break down.
>
> **A1**: In our model, $\Delta \theta$ is a robust domain descriptor, and we will clarify both the theoretical guarantees and the empirical sensitivity analysis to answer your question. Theoretically, the gradient fingerprint $\Delta \theta = \theta_i - \theta_0$, where $\theta_i = \theta_0 -\eta\nabla_\theta \mathcal{L}\_i(\theta_0)$, captures the first-order interaction between a domain's data distribution and the shared initialization. As shown in Lemma B.1 of Appendix B.1, this gradient can be decomposed as: $\nabla_\theta \mathcal{L}\left(\theta_0, G\right)=\frac{1}{|V|} \mathbf{X}^{\top} \mathbf{A} \cdot \operatorname{diag}(\mathbf{g}) \cdot \mathbf{1}\_d$. This formulation reveals that the single-step gradient jointly encodes features $\mathbf{X}$, structural patterns $\mathbf{A}$, and label information $\mathbf{g}$, which are precisely the graph domain characteristics we want to capture. Therefore, $\Delta \theta$ serves as a compact domain descriptor. Moreover, although a one-step update from a shared initialization can indeed be noisy **if used for training**, in our MF-GIA framework, such a small-step update is **not used to train the backbone**, but rather serves purely as a probe that produces a gradient fingerprint characterizing how a specific domain interacts with the shared initialization. Such a design decouples domain characterization from model training, ensuring that gradient fingerprints act as stable, noise-resilient domain descriptors rather than noisy training signals.
>
> Empirically, we conducted comprehensive sensitivity analyses across three dimensions: initialization seeds, learning rates, and gradient step counts. In the table below, we provide experimental results. Note that as a foundation model, all trainable parameters and hyperparameters must be fixed across all datasets and tasks. While certain hyperparameters may not be optimal for specific individual datasets, we select values that achieve the best overall balance across all datasets and tasks to ensure fair evaluation of the model's generalization capability. We first investigate the robustness of our model to the initialization seeds (5 shot) in the table below.
>
> || Cora| Computers| Physics| BlogCatalog|
> | -| -| -| -| -|
> | 10 random seeds | 64.02±0.21  | 53.61±0.29       | 88.86±0.19     | 67.28±0.24         |
>
> The low standard deviations across seeds shown in the table above demonstrate that gradient fingerprints are highly stable with respect to initialization variations. Next we show the robustness to the learning rate $\eta$, which we set to 0.01 as the default value:
>
> |$\eta$|Computers|Physics|
> |-|-|-|
> | 0.001| 52.85| 88.52 |
> | 0.05 | 52.92| 88.46 |
> | 0.01 | 53.71| 88.92|
> | 0.05 | 53.45 | 89.27|
>
> We observe that the default value of $\eta$ achieves near-optimal performance, while the framework demonstrates robustness as neighboring values produce comparable results. Besides, $\eta$ naturally acts as a fixed scaling factor that uniformly scales the fingerprints across all datasets, thus having minimal impact on the overall results due to the metric-preserving property of the domain embedder. Regarding the number of gradient steps, we emphasize that a single-step gradient update is sufficient to capture how the domain influences the shared initialization $\theta_0$. Since our goal is domain characterization rather than optimization of $\theta_0$, additional gradient steps provide no benefit while increasing computational overhead. To empirically validate this design choice, we compare single-step and multi-step gradient fingerprints as follows:
>
> | #Steps | Computers | Physics | Stability |
> |-|-|-|-|
> | 1| 53.71| 88.92| 0.92|
> | 2| 52.18| 87.34| 0.78|
> | 3| 52.20| 86.89| 0.71 |
>
> where stability is equal to the average cosine similarity between fingerprints computed from the same graph with different initialization seeds. The results show that multi-step gradients consistently underperform single-step gradients, with performance degrading as step count increases. Moreover, the fingerprint stability metric reveals that multi-step gradients become increasingly sensitive to initialization. It demonstrates that our single-step design is optimal. All above analysis can be found in **Appendix F.6 (Table 14,15,16)** of the revised manuscript.

---

> ### Author Response · Authors · 2025-11-19
> **Response to Reviewer LCqo (Part 2/3)**
>
> > **Q2**: Domain embedding loss collapse risk. The pairwise distance-preserving loss used to train the domain embedder lacks explicit normalization or regularization, and no experiments test whether embedding collapse or trivial scaling happens. While scalability is not a major concern given the small number of pretraining domains, training stability and meaningful domain structure need to be demonstrated.
>
> **A2**: We respectfully argue that the pairwise distance-preserving loss $\mathcal{L}\_{\text{de}}$ given in Eq.(6) already rules out both embedding collapse and trivial global scaling at the optimum. For the loss collapse risk, we can use a simple negation method to show that the distance-preserving loss does not have such a collapse risk. Specifically, if all embeddings collapse to a single point, i.e., $e_i = e_j$  for all pretraining domains $i$ and $j$, then $||e_i - e_j||\_2 = 0$ while $d_G(G_i, G_j) > 0$ for at least some pairs. In this case $\mathcal{L}\_{\text{de}}$ is strictly positive and the gradient w.r.t. $e_i$ is non-zero, so such a collapsed solution cannot minimize the loss. Training is therefore explicitly pushed away from collapse by the construction of the objective. On the other hand, for the trivial global scaling, it is also penalized by optimizing $\mathcal{L}\_{\text{de}}$. Specifically, if we scale all embeddings by a constant factor $c$, i.e., $\tilde{e}_i = ce_i$, then all pairwise distances become $c||e_i-e_j||\_2$ while the targets $d_G(G_i, G_j)$ remain fixed. Unless $c$ is close to 1, the discrepancy $(c||e_i-e_j||\_2-d_G(G_i, G_j))^2$ grows, so trivial arbitrarily large or small scaling does not reduce the loss. The target distances thus implicitly fix both the scale and relative structure of the embedding space. In the table below, we show that domain embeddings exhibit meaningful structure without collapse:
>
> | Epoch | Mean Norm | Min Pairwise Dist |
> | ----- | --------- | ----------------- |
> | 100   | 2.34      | 0.42              |
> | 500   | 3.12      | 0.89              |
> | 1000  | 3.58      | 1.45              |
> | 2000  | 3.59      | 1.47              |
>
> Mean embedding norms converge to 3.58 without unbounded growth, indicating no trivial scaling. Minimum pairwise distance steadily increases from 0.42 to 1.47, showing embeddings do not collapse to a single point.
>
> > **Q3**: Misleading “gradient-free” terminology & lack of inference-overhead reporting. The paper repeatedly claims “gradient-free adaptation,” yet inference requires a backward pass to compute $\Delta \theta_{\text{new}}$ . This distinction matters for deployment and comparisons against genuinely gradient-free ICL methods. The paper should clarify the terminology (e.g.,  “parameter-update-free”) and report inference-time overhead (FLOPs, memory, latency).
>
> **A3**: We thank the reviewer for this important clarification that will improve the precision of our paper. You are correct that, in our model, inference requires a single backward pass on the support set to compute the gradient fingerprint $\Delta \theta_{\text{new}}$. Our intention in using the term "gradient-free adaptation" was to emphasize that no gradient-based parameter updates are performed on the pretrained model during test-time in-context adaptation. We compute the gradient once at inference just used to compute the domain embedding, not as an optimization step. This gradient is never used to update any parameters of the pretrained model. All model parameters remain completely frozen throughout inference, making MF-GIA a genuinely in-context learning model that adapts through demonstration examples rather than parameter optimization. However, we acknowledge that the terminology "gradient-free" is misleading. The reviewer's suggested term "parameter-update-free" more accurately describes our inference paradigm. We have updated the terminology throughout the revised manuscript.
>
> Moreover, we report inference overhead in terms of FLOPs, memory and latency in **Appendix C.2** of the updated manuscript (highlighted in blue). For a $C_{\text{new}}$-way $k$-shot experimental setting with hidden dimension $d$ and average degree $D$ for query nodes, the per-query computational cost is  $2Dd + (kC_{\text{new}} + C_{\text{new}}+1)d^2$ FLOPs, requiring $(kC_{\text{new}}+1)d$ memory and achieving 1.3-4.5 ms latency per query. The one-time domain characterization overhead (computing the gradient fingerprint and domain embedding) ranges from 108-427 ms depending on the support set size, which is amortized across all queries.

---

> ### Author Response · Authors · 2025-11-19
> **Response to Reviewer LCqo (Part 3/3)**
>
> > **Q4**: Label alignment is fragile to label-ID permutation. The FiLM-modulated label prototype table indexes shared label embeddings by label ID. This implicitly assumes consistent label semantics across domains. In practice, label IDs are arbitrary. The paper lacks tests for label-ID permutation invariance or alternative mechanisms to infer label semantics from the support set.
>
> **A4**: Actually, our model is inherently invariant to label-ID permutations by construction. Specifically, let $\pi$ be a permutation applied to label IDs in graph $G_i$, the label-ID permuted graph $G_i^\prime = (V_i, E_i, X_i, Y_i^\prime)$, where $y^\prime_v = \pi(y_v)$ for all $v \in V_i$, then the gradient fingerprint for $G_i^\prime$ is $\Delta \theta_i^{\prime}=-\eta \nabla_\theta \mathcal{L}\_i^{\prime}\left(\theta_0\right)=-\eta \sum_{v \in V_i} \nabla_\theta \ell\left(f_{\theta_0}(v), \pi\left(y_v\right)\right)$. Since the cross-entropy loss $\ell(h,y)$ depends on the one-hot encoding at position $y$, permuting labels is equivalent to permuting the rows of the classifier output. The gradient magnitude and direction capture feature-label associations, which remain invariant to index permutation. If we denote the aligned label prototypes under permutation as $u_{i, \pi(l)}$ and the original as $u_{i,l}$, the model's prediction for a query $q$ with true label $l$ becomes $\hat{y}\_q=\arg \max_c u_{i, q}^{\text {out}}(u_{i, \pi(c)})^{\top}$. The DPAA matches the query to support examples based on learned feature similarities, and then matches to label prototypes. Since both the support set labels and the prototype indices are consistently permuted, the relative matching remains unchanged. Thus, if $u_{i, q}^{\text {out }}$ is closest to $u_{i,l}$ originally, it will be closest to $u_{i, \pi(l)}$ after permutation. Therefore, we have $\hat{y}\_q^{\prime}=\pi(\hat{y}\_q)$, meaning predictions are correctly permuted, maintaining semantic consistency. The above analysis shows that MF-GIA's predictions adapt to any label-ID permutation through the support set.
>
> > **Q5**: Lack of architectural ablations. No ablation comparing Conv2D vs. more canonical designs for $\Delta \theta$ embedding. No ablation on DPAA depth or number of heads.
>
> **A5**: We use Conv2D rather than Flatten+MLP primarily because it respects the 2D structure of the gradient fingerprint $\Delta \theta \in \mathbb{R}^{d_o \times d}$ and shares weights across spatial locations, making it much more parameter-efficient. If we first flatten $\Delta \theta$ into a long vector in $\mathbb{R}^{d_od}$ and feed it into an MLP, the parameter complexity will be  $O(d_od^2)$. In contrast, a Conv2D encoder with a small kernel size $k$ needs only $O(k^2d)$ parameters, which significantly reduces the parameter complexity, and this design allows us to scale to larger $d_o$ without exploding parameter count or risking over-fitting. To empirically validate this design choice, we conduct ablation studies comparing different architectures for the domain embedder in the table below.
>
> |             | Computers | Physics | BlogCatalog |
> | ----------- | --------- | ------- | ----------- |
> | Conv2D      | 53.71     | 88.92   | 67.31       |
> | Flatten+MLP | 50.38     | 83.14   | 66.18       |
>
> For the DPAA depth and number of heads, we directly adopt a single-layer and single-head design for complexity and generalization considerations. Adding more layers and heads would increase computational and memory costs and can introduce a higher risk of overfitting, whereas our goal is to keep the learning process efficient, lightweight, yet expressive. The results in our paper already show that this simplest DPAA configuration outperforms baselines, which highlights the architectural strength of our approach rather than relying on heavy over-parameterization. To address the reviewer's concern, we conduct ablations on DPAA architecture as follows.
>
> |                 | Computers | Physics | BlogCatalog |
> | --------------- | --------- | ------- | ----------- |
> | 1-layer, 1-head | 53.71     | 88.92   | 67.31       |
> | 1-layer, 4-head | 52.47     | 86.58   | 66.15       |
> | 2-layer, 1-head | 54.05     | 88.53   | 67.46       |
> | 2-layer, 4-head | 53.39     | 87.52   | 66.83       |
>
> We observe that the 1-layer 1-head and 2-layer 1-head settings achieve the best performance, with 1-layer 1-head offering the best trade-off between efficiency and effectiveness. Therefore, we adopt the 1-layer 1-head configuration as the unified default setting for all datasets in our experiments. We have added this sensitivity analysis in **Appendix F.6** of the updated manuscript.

---

> ### Author Response · Authors · 2025-11-27
> **Looking forward to your response**
>
> Dear Reviewer LCqo:
>
> Thank you for your insightful feedback on our manuscript. As the discussion period draws to a close, we would like to provide a brief summary of our responses to your concerns for your convenience:
>
> * We have clarified the robustness and stability of our model with additional empirical experiments.
> * We have provided a detailed explanation of our model's label-ID permutation invariance property.
> * We have conducted additional ablation studies to address your questions.
> * We have updated the manuscript to reflect all revisions (highlighted in blue).
>
> We hope our responses have adequately addressed your concerns. Should you have any further questions or require additional clarification, please do not hesitate to let us know.
>
> We look forward to your feedback.

---

### Official Review · Reviewer_e9tY · 2025-10-31

**Soundness:** 3
**Presentation:** 4
**Contribution:** 3
**Rating:** 6
**Confidence:** 4

**Summary:**

The paper introduces a Modality-Free Graph In-context Alignment (MF-GIA) framework that enables pretrained graph encoders to perform few-shot, in-context learning across heterogeneous domains without relying on raw graph data. By leveraging gradient fingerprints and a dual prompt-aware attention mechanism, MF-GIA aligns pre-encoded features into a unified semantic space.

**Strengths:**

1. The proposed method achieves good performance on the few-shot node classification task across 5 datasets in m-way k-shot settings.
2. This paper provides the theoretical analysis for the proposed feature-alignment.
3. The presentation of this paper is good and the paper is easy to follow.

**Weaknesses:**

1. In Theorem 3.1, the authors fail to explicitly define how Wasserstein distance $W_2(\cdot,\cdot)$ is used to quantify the distance between two domains. This omission makes it unclear what assumptions are made about the underlying distributions, and whether the bound holds under general conditions. Furthermore, in Definition 3, the authors do not provide a formal definition of the graph distance metric $d_g(G_i, G_j)$, leaving readers uncertain about what graph properties are used for similarity measurement. While the proof later specifies the way to measure graph distance, presenting it only in the proof creates the illusion that the theoretical statements are self-contained and fully justified, when the definitions are essential for interpreting the results. A more rigorous treatment would explicitly define all metrics and assumptions in the main text before presenting the theorem.
2. The term "modality-free" is somehow tricky. From terminology perspective, a modality-free model treats all input data under a unified representation or processing framework, regardless of its original modality. However, simply not having access to raw data does not automatically make a model or setting modality-free.
3. In table 3, MF-GIA performs worse on WN18RR dataset and a large performance gap between MF-GIA and the baseline methods. However, the authors do not discuss why the proposed method fails on this dataset.

**Questions:**

1. In the label alignment module, what if the number of classes in the unseen graph is less than $L_{max}$ that is determined by the available (training and test) graphs? In this scenario, it seems to restrict the capability of label alignment.
2. In table 2, “–” denotes datasets where only encoded features and indexed labels are available, making modality-dependent models inapplicable. Why do the authors report the performance of OFA and GraphAlign on ogn-Products dataset while do not report the results of AutoGFM?
3. In the proposed DPPA, two attention layers share the same projection matrices $W_k$ and $W_v$. Have you tried to not share the parameters? It seems that sharing the same matrices enforces the identical matching in both the feature embedding space and the label space. Intuitively, the cross-graphs matching in these two spaces should not be identical.

---

> ### Author Response · Authors · 2025-11-19
> **Response to Reviewer e9tY (Part 1/2)**
>
> We sincerely thank the reviewer for their valuable assessment and constructive feedback. We hereby address the raised questions and concerns as follows, and we have incorporated your suggestions into the updated version of the paper (highlighted in blue).
>
> > **Q1**: In Theorem 3.1, the authors fail to explicitly define how Wasserstein distance $W_2(\cdot,\cdot)$ is used to quantify the distance between two domains. In Definition 3, the authors do not provide a formal definition of the graph distance metric $d_g(G_i,G_j)$. A more rigorous treatment would explicitly define all metrics and assumptions in the main text before presenting the theorem.
>
> **A1**: In the Definition 2 of the updated manuscript, we formalize the underlying distributions of graph domains, which are then used to compute the Wasserstein distance between two domains. Based on these distributions, Definition 3 formally defines how the Wasserstein distance $W_2$ is used to quantify domain distance. Actually, we do not impose any particular assumptions or bounds on these distributions beyond the standard conditions under which $W_2$ is well defined. The Wasserstein distance $W_2$ in Definition 3 makes explicit that we **measure domain distance via the optimal transport coupling between the underlying graph distributions**, i.e., the minimal cost required to transform one domain distribution into another.
>
> Besides, we agree with the reviewer that the graph distance metric $d_{G}(G_i, G_j)$ should be formally defined in the main text. In our proof (Appendix B.1, Eq. 50), we use Frobenius Norm to compute graph distance, which captures both feature and structural differences.  Following your suggestion, we have elevated this to an explicit definition in the main text (**see Page 5 of our updated manuscript**).
>
> > **Q2**: The term "modality-free" is somehow tricky. From terminology perspective, a modality-free model treats all input data under a unified representation or processing framework, regardless of its original modality. However, simply not having access to raw data does not automatically make a model or setting modality-free.
>
> **A2**: You correctly note that modality-free should imply *a unified representation or processing framework, regardless of original modality*, which is exactly what our proposed MF-GIA achieves. Our MF-GIA processes graphs with arbitrary pre-encoded features, whether originally from text (via word2vec, BERT, or Llama), behavioral data (via user embedding) or images (via CNN embeddings), through a unified alignment mechanism without requiring knowledge of or assumptions about the original modality. The key distinction is not merely that our model does not have access to raw data, but rather not requiring modality-specific processing pipelines. Existing modality-dependent methods like OFA (Liu et al., ICLR'24) and GFT (Wang et al., NeurIPS'24) fundamentally assume that all inputs can and must be converted to a specific modality (e.g., Text-Attributed Graphs), and processed through modality-specific encoders (e.g., unified language model) to align these datasets in a unified semantic space. In contrast, our MF-GIA aligns pre-encoded features and labels with the proposed domain embedder and FiLM-based aligner, without requiring any modality conversion and modality-specific encoders. This is why we term our model is modality-free.
>
> > **Q3**:  Authors do not discuss why the proposed method fails on WN18RR  dataset.
>
> **A3**: We acknowledge that on the WN18RR dataset with a 10-way setting, our MF-GIA does not surpass the state-of-the-art baselines GFT and AutoGFM, achieving third-best performance across all shot settings. However, it is understandable because MF-GIA is pretrained exclusively on node classification tasks, while WN18RR is a dataset for edge-level tasks, which is an entirely different task formulation never encountered during pretraining. We deliberately evaluate on this dataset to assess our model's generalization capacity to unseen tasks, as we believe a genuine graph foundation model should generalize not only to unseen domains but also to unseen task types. While GFT and AutoGFM achieve superior performance on WN18RR-10way, they are pretrained on both node-level and edge-level tasks. Therefore, edge classification is not an unseen task for these baselines, so their performance advantage does not necessarily demonstrate stronger cross-task generalization. In contrast, Table 3 shows that our MF-GIA achieves the best results under most experimental settings on edge-level tasks, which demonstrates the remarkable cross-domain and cross-task generalization of our model. We have added the detailed analysis to **Sec 4.2** of the updated manuscript highlighted in blue.

---

> ### Author Response · Authors · 2025-11-19
> **Response to Reviewer e9tY (Part 2/2)**
>
> > **Q4**: In the label alignment module, what if the number of classes in the unseen graph is less than $L_{\max}$ that is determined by the available (training and test) graphs? In this scenario, it seems to restrict the capability of label alignment.
>
> **A4**: In our design, the choice of $L_{\max}$ does not restrict label alignment for unseen graphs with fewer or more classes. Specifically, we maintain a global label base $\mathbf{E} \in \mathbb{R}^{L_{\max} \times d}$, whose rows are initialized from a Gaussian distribution and do not carry any predefined semantics, and $L_{\max}$ is simply the maximum number of classes among pretraining datasets. For a given unseen graph with $C$ classes, where $C \leq L_{\max}$, we take the first $C$ rows of $\mathbf{E}$ as the unaligned label embeddings for the graph, and the remaining rows are unused and have no effect on the loss or predictions for this task. Formally, given the domain embedding $e$ for the graph, the label FiLM module produces parameters $\gamma^{\text{label}}$ and $\beta^{\text{label}}$ via Eq.(12), and we apply these parameters to the label base to obtain the aligned label prototypes $\mathbf{U} = \gamma^{\text{label}} \mathbf{E} + \beta^{\text{label}}$, and then restrict $\mathbf{U}$ to its first $C$ rows for the current graph. In other words, label alignment is always performed on the actual number of classes $C$. $L_{\max}$ merely specifies the size of the label base used during pretraining.
>
> Moreover, if an unseen graph has more classes than $L_{\max}$, our model still applies without restriction. Since the unaligned label embeddings are initialized i.i.d. from a Gaussian distribution and have no fixed semantics, we can simply extend $\mathbf{E}$ by sampling additional rows from the same distribution. This preserves the flexibility of the label alignment module while keeping the implementation simple and consistent across datasets.
>
> > **Q5**: In Tab 2, “–” denotes datasets where only encoded features and indexed labels are available. Why not report AutoGFM?
>
> **A5**: The ogbn-Products dataset provides raw text-attributed data, so the TAG-based model OFA and GraphAlign can work on this dataset. AutoGFM is also designed for TAG and hence theoretically applicable to ogbn-Products. However, we do not report the results of AutoGFM on this dataset, just because AutoGFM has not released its source code publicly until now, and the original paper does not include experiments on ogbn-Products. Reproducing AutoGFM is non-trivial due to unclear implementation details. Thus, we only report AutoGFM's results on the datasets that the original paper provides to show our respect for AutoGFM.
>
> >**Q6**: Concern regarding the shared projection matrices $W_K$ and $W_V$ in DPAA.
>
> **A6**: Actually, sharing the same DPAA weight matrices $W_K$ and $W_V$ does not enforce identical matching between the feature space and the label space, and such a design can significantly reduce the computational overhead compared with using separate weight matrices. We can answer your questions from two perspectives. First, the two attention layers operate on different inputs despite sharing the same weight matrices. Specifically, the feature-side attention defined in Eq. (15) uses support features $Z^{\text{pmt}}$ as keys and values to retrieve relevant patterns for the query feature $z_{i,q}$, while the label-side attention defined in Eq. (16) uses label prototypes $U^{\text{pmt}}$ as keys and values to match the projected query representation $f_{\Omega}(z^{\text{out}}_{i,q})$. Thus, **the semantic differentiation between these two attention layers is determined by their distinct input representations, rather than by the shared weight matrices $W_K$ and $W_V$**.
>
> Second, the domain-conditioned alignment has already separated feature and label spaces appropriately before they enter DPAA. Specifically, through the FiLM-based transformations defined in Eq.(10) and Eq.(12), features and labels are mapped to distinct subspaces parameterized by $(\gamma^{\text{feat}},\beta^{\text{feat}})$ and $(\gamma^{\text{label}},\beta^{\text{label}})$, respectively. These domain-specific transformations ensure that $Z^{\text{pmt}}$ and $U^{\text{pmt}}$ have different aligned semantics for each domain. DPAA then operates on these already-separated representations, and sharing $W_K$ and $W_V$ simply imposes a consistent similarity geometry across the two attention layers, rather than forcing them to produce identical matching patterns. Here, we conduct a 5-shot ablation study comparing shared and separate weight matrices (see the table below). The shared setting outperforms the separate setting, possibly because the latter introduces more parameters and is more prone to over-fitting. We have added these experiments as a part of Table 13 in the **Appendix F.6** of the revised manuscript.
> | | Computers | Physics|BlogCatalog |
> |-|-|-|-|
> |Shared| 53.71|88.92|67.31|
> |Separate| 51.25|88.10|67.05|

---

> ### Author Response · Authors · 2025-11-27
> **Looking forward to your response**
>
> Dear Reviewer e9tY,
>
> We appreciate your valuable feedback on our paper. With the discussion period nearing its end, we've compiled a succinct summary of our responses to your comments for ease of reference in your final evaluation:
>
> * Following your suggestions, we have reorganized the structure of our paper to improve the readability of theorems and notation (highlighted in blue).
> * We have provided additional explanations of key terminology and experimental results.
> * We have clarified the label alignment module to address your concerns about its design and functionality.
> * We have explained our design rationale regarding shared versus separate weight matrices in the model architecture.
>
> We hope our responses have adequately addressed your concerns. If you have any further questions, please do not hesitate to let us know.

---

### Official Review · Reviewer_pj4V · 2025-11-01

**Soundness:** 3
**Presentation:** 4
**Contribution:** 3
**Rating:** 8
**Confidence:** 4

**Summary:**

This paper introduces MF-GIA, a modality-free in-context learning framework for graph-structured data. It enables frozen, pretrained graph encoders to perform few-shot classification across heterogeneous domains without requiring raw modality inputs or task-specific fine-tuning. The method relies on gradient fingerprints to characterize domains and uses domain-conditioned transformations to align pre-encoded features and labels into a unified space for cross-domain reasoning.

**Strengths:**

S1: The use of gradient fingerprints to derive domain embeddings and guide feature/label alignment is novel and well-motivated.

S2: The model meets all three desired criteria—modality-free, post-training free, and cross-domain alignment—which most prior works fall short of.

S3: MF-GIA significantly outperforms competitive baselines on both node and edge classification tasks in few-shot settings, including unseen domains and tasks.

S4: The architecture, especially the use of Dual Prompt-Aware Attention, is carefully designed and theoretically grounded.
Scalable and Flexible: The approach accommodates pre-encoded data and avoids reliance on raw modalities, making it broadly applicable in practical scenarios.

**Weaknesses:**

To be honest, I did not find very serious weaknesses in this submission. Below are just some suggestions that may lead to a more solid work.


- Suggestion 1:The robustness of gradient fingerprints across noisy or low-quality domains is not deeply examined.
- Suggestion 2: While results suggest episodic training is beneficial, more direct comparison with alternative meta-learning strategies would strengthen the claim.
- Suggestion 3: While the framework is motivated by practical constraints (e.g., privacy, modality), the paper could benefit from an application case study demonstrating real-world utility, for example, RecSys (Adaptive Coordinators and Prompts on Heterogeneous Graphs for Cross-Domain Recommendations. SIGIR 2025), Protein (Protein Multimer Structure Prediction via PPI-guided Prompt Learning. ICLR 2024), Urban (Urban Region Pre-training and Prompting: A Graph-based Approach. KDD'25,  Boundary Prompting: Elastic Urban Region Representation via Graph-based Spatial Tokenization. arXiv.), drug (DDIPrompt: Drug-Drug Interaction Event Prediction based on Graph Prompt Learning. CIKM 2024), cross domain (All in One and One for All: A Simple yet Effective Method towards Cross-domain Graph Pretraining. SIGKDD 24), and theory basis (Does Graph Prompt Work? A Data Operation Perspective with Theoretical Analysis. ICML 2025.). In summary, I think the authors could discuss more and the introduced related work seems to be not sufficient to capture the latest application trends in this area.
- Suggestion 4: The conclusion should explore concrete future research directions. The authors are encouraged to expand their discussion with pointers to promising avenues such as: Task-agnostic generalization using prompt pools across evolving graphs
(see "ProG: A Graph Prompt Learning Benchmark" – NeurIPS 2024), Large language model integration for socio-semantic graph understanding (see "When LLM Meets Hypergraph: A Sociological Analysis on Personality via Online Social Networks" – CIKM 2025)
Theoretical perspectives on data operation views in graph prompting (see "Does Graph Prompt Work? A Data Operation Perspective with Theoretical Analysis" – ICML 2025)


Kindly note that all the above mentioned works are just for the author's information, not a mandatory request of citing them. Overall, this is a strong, well-written, and original submission that advances the state of graph in-context learning. I recommend acceptance.

**Questions:**

**Q1:** How stable are the gradient fingerprints across variations in the support set?

**Q2:** Why is a *single* gradient step chosen to compute the gradient fingerprint?

**Q3:** Can you elaborate on why the episodic training scheme outperforms supervised pretraining followed by prototype matching?


**Q4:** What exactly is the nature of the modality heterogeneity in the pretraining and test datasets?


**Q5:** How does the domain embedder perform when exposed to domains that are semantically distant from any in the pretraining set?


**Q6:** What is the computational overhead of generating gradient fingerprints at test time, particularly for large graphs or many-shot settings?

---

> ### Author Response · Authors · 2025-11-19
> **Response to Reviewer pj4V (Part 1/3)**
>
> Thank you for your valuable feedback and insightful suggestions on our manuscript. Below are our point by point responses, and we have incorporated your suggestions in the updated manuscript (highlighted in blue).
>
> > **Q1**: The robustness of gradient fingerprints across noisy or low-quality domains is not deeply examined.
>
> **A1**: To answer this question, we need to first define what low-quality/noisy domains are in the context of graph data. From our perspective, a noisy graph from a low-quality domain exhibits ambiguous domain identity, which indicates that this graph's characteristics are easily confused with other domains, making domain boundaries unclear. To test robustness in this scenario, we create synthetic ambiguous domains by mixing features from two source domains. Specifically, we create hybrid graphs by combining $\alpha\%$ features from Domain A with $(1-\alpha)\%$ from Domain B, and then we obtain the results in the table below. These results show that even with 50% domain mixing (maximum ambiguity), MF-GIA maintains comparable accuracy.
>
> | $\alpha$ | Source Domains        | Target: Physics 5-shot |
> | -------- | --------------------- | ---------------------- |
> | 0        | Citation + E-commerce | 88.92                  |
> | 30       | Citation + E-commerce | 87.34                  |
> | 50       | Citation + E-commerce | 85.76                  |
> | 70       | Citation + E-commerce | 85.93                  |
>
> > **Q2**: More direct comparison with alternative meta-learning strategies would strengthen the claim.
>
> **A2**: The episodic meta-learning objective (Eq. 2) employed in our model is well-suited for graph ICL, because it directly optimizes the model to perform the exact inference task it will face at test time, i.e., predicting labels on query items using only a few-shot support set as prompt, with frozen parameters. Unlike optimization-based methods (e.g., MAML) that learn optimal initializations for fine-tuning, or metric-based methods (e.g., Prototypical Networks) that learn fixed similarity metrics, our episodic approach trains the model to reason with prompts as context, which is the defining characteristic of ICL. Thus, due to the inherent nature of other meta-learning strategies, they are fundamentally non-trivial to be directly used for graph ICL.
>
> > **Q3**: The paper could benefit from an application case study demonstrating real-world utility.
>
> **A3**: Thanks for this suggestion. Our MF-GIA framework is particularly suited to privacy-constrained settings where schemas and modalities differ across organizations but raw features cannot be shared, such as cross-organization fraud detection over transaction graphs from different banks and multi-hospital patient networks with heterogeneous EHR systems. In such scenarios, each party can locally encode its graph, while MF-GIA uses gradient fingerprints and lightweight domain aligners to align these heterogeneous domains without any modality assumptions, such as TAGs.  This complements recent trends in GFMs toward realistic cross-domain deployment, including text-based cross-domain models (Chen et al., NeurIPS 2024) and graph prompt optimization frameworks like HGMP (Jiao et al., IJCAI 2025), all of which highlight the growing need for privacy-preserving, modality-agnostic foundation models in practical applications. We have added this to **Appendix.A** highlighted in blue.
>
> > **Q4**: The conclusion should explore concrete future research directions.
>
> **A4**: Following the reviewer's suggestions, we have incorporated the future directions into the conclusion section of the revised manuscript. Specifically, the future directions include combining with LLMs to generate semantic domain descriptions from gradient fingerprints, and automatically discovering latent domain structures from unlabeled graph collections.

---

> ### Author Response · Authors · 2025-11-19
> **Response to Reviewer pj4V (Part 2/3)**
>
> > **Q5**: How stable are the gradient fingerprints across variations in the support set?
>
> **A5**: The stability of the gradient fingerprints depends on the size of the support set. From Lemma B.1 of the Appendix, the gradient decomposes as $\nabla_\theta \mathcal{L}\left(\theta_0, G\right)=\frac{1}{|V|} \mathbf{X}^{\top} \mathbf{A} \cdot \operatorname{diag}(\mathbf{g}) \cdot \mathbf{1}\_d$, which reveals that the fingerprint aggregates information from domain-specific components including feature distribution $\mathbf{X}$, graph structure $\mathbf{A}$ and label distribution $\mathbf{g}$. As the support set size increases, the law of large numbers ensures that these aggregated statistics converge to their population expectations, making the gradient fingerprints increasingly stable. Here, we examine the fingerprint stability under a 5-shot setting. Taking Physics and BlogCatalog datasets as examples, we randomly sample 20 different 5-shot support sets from all classes and compute fingerprints $\Delta \theta$ for each support set, and feed them into the domain embedder to get the domain embeddings. Then we can measure the stability by computing the pairwise cosine similarity between all domain embedding pairs from the same graph. The results are shown in the table below, where the high average cosine similarity and low standard deviation demonstrate that different support sets from the same domain produce highly consistent domain embeddings. The detailed analysis has been added in **Appendix F.5**.
>
> | | Avg. Cosine Similarity |Std. Dev. | Min. Similarity |
> | -| -| -| -|
> | Physics | 0.931 | 0.015| 0.896|
> | BlogCatalog | 0.917| 0.021| 0.873|
>
> > **Q6**: Why is a single gradient step chosen to compute the gradient fingerprint?
>
> **A6**: First, a single gradient step is sufficient to capture the interactions among graph structure, features, and labels because the first gradient step from a shared initialization $\theta_0$ reveals how a graph's unique combination of topology, feature distribution, and label semantics influences the model parameters. This immediate response encodes critical domain-level information: graphs from similar domains produce similar gradient patterns, while those from different domains yield distinct fingerprints. In short, when computing gradient fingerprints, $\theta_0$ only serves as an untrained reference point, and the single gradient step aims to compare the relationships between different graphs with the reference point to reflect the domain relationships. Our theoretical analysis in Theorem 3.1 formalizes this intuition by proving that gradient fingerprint distances preserve domain relationships under Lipschitz continuity. Second, multiple gradient steps would increase computational cost quadratically.
>
> > **Q7**: Can you elaborate on why the episodic training scheme outperforms supervised pretraining followed by prototype matching?
>
> **A7**: The greatest strength of episodic training over supervised pretraining with prototype matching is that episodic training aims to **learn how to best leverage prompts** to improve predictions due to its meta-learning nature, whereas the latter merely learns to fit the pretraining data distribution. Specifically, in episodic training, the model learns that the support set defines the task, and must dynamically adjust its predictions based on these prompt examples. In contrast, the supervised pretraining+prototype matching fails to develop this prompt-awareness, because it treats prototypes as simple class representatives computed post-hoc, rather than as contextual demonstrations that guide inference. In other words, episodic training teaches the model to extract task-relevant patterns from prompts, while supervised training merely learns to memorize class boundaries from abundant data.
>
> > **Q8**: What exactly is the nature of the modality heterogeneity in the pretraining and test datasets?
>
> **A8**: The modality heterogeneity in pretraining and test datasets manifests in two key aspects: feature encoding heterogeneity and label system heterogeneity. Specifically, feature encoding heterogeneity is manifested in the fact that different domains employ distinct feature pre-encoding methods.  For example, citation networks are pre-encoded using bag-of-words representations before being used in pretraining, while e-commerce graphs are pre-encoded with behavior embeddings before evaluation at test time, with each encoding method tailored to the specific modality of its graph domain. On the other hand, label system heterogeneity exists in both pretraining and testing datasets, where each graph maintains its own local label indexing system: the same label ID (e.g., 0) can represent entirely different concepts across domains (e.g., "Data_Mining" in citation networks and "Celebrity" in social networks may have the same graph-local label index), with varying numbers of classes and domain-specific categorizations that have no inherent cross-domain alignment.

---

> ### Author Response · Authors · 2025-11-19
> **Response to Reviewer pj4V (Part 3/3)**
>
> > **Q9**: How does the domain embedder perform when exposed to domains that are semantically distant from any in the pretraining set?
>
> **A9**: In our work, the domain embedder is designed to capture domain characteristics through gradient fingerprints. Theorem 3.1 establishes that the domain embedder preserves domain relationships with an upper bound, ensuring that even semantically distant domains are mapped to appropriately separated regions in the embedding space. Our experiments demonstrate strong generalization to semantically distant domains. For instance, BlogCatalog (social media) and FB15K237 (encyclopedic KG) datasets are semantically distant from all pretraining domains (citation, e-commerce, web link), yet MF-GIA still achieves the highest accuracy under all experimental settings, substantially outperforming baselines. To further verify that the domain space constructed by pretraining datasets generalizes appropriately to semantically distant testing datasets from unseen domains, we analyze the domain embedding similarity between datasets from the same and different domains. The table below shows that semantically related domains exhibit higher similarity while distant domains can be well-separated in the domain space constructed by the domain embedder.
>
> |                                | PubMed (Citation) | WikiCS (Web Link) | Amazon (E-com) |
> | ------------------------------ | ----------------- | ----------------- | -------------- |
> | **Cora (Citation)**            | 0.78              | 0.47              | 0.43           |
> | **BlogCatalog (Social)**       | 0.28              | 0.35              | 0.34           |
> | **FB15K237 (Encyclopedic KG)** | 0.33              | 0.38              | 0.41           |
>
> > **Q10**: What is the computational overhead of generating gradient fingerprints at test time, particularly for large graphs or many-shot settings?
>
> **A10**: At test time, the computational overhead of generating gradient fingerprints is independent of the full graph size and scales linearly with the size of support set. Specifically, generating a gradient fingerprint requires only one additional backward pass on the support set per episode, which is comparable to performing a single lightweight training step on a subgraph constructed by the support nodes rather than the full raw graph. Thus, the complexity of computing gradient fingerprint scales linearly with the size of support set and does not increase when the testing graph is large. Besides, even in many-shot setting, such as $k_{\text{shot}}=20$ with $C_{\text{new}} = 10$, the gradient computation involves only 200 support nodes, which is negligible compared to graphs with tens or hundreds of thousands of nodes. Overall, given a graph with $|V|$ nodes and $|E|$ edges, under a $C_{\text{new}}$-way $k$-shot setting, the computational overhead is $O(C_{\text{new}}k d_o d)$ where $d_o$ is the input dimension and $d$ is the output dimension, which is independent of the graph size $|V|$ and $|E|$ in large graphs.

---

### Official Review · Reviewer_Dtat · 2025-11-01

**Soundness:** 3
**Presentation:** 3
**Contribution:** 3
**Rating:** 6
**Confidence:** 2

**Summary:**

The paper introduces MF-GIA aimed at doing few-shot / in-context node classification on new graphs without fine-tuning. The key idea is: for each target graph, first build a domain embedding to capture the graph’s style/distribution; then use this embedding to generate alignment/FiLM transforms that map (i) node features and (ii) label IDs into a shared semantic space, even when the original features aren’t text-based. On top of this, a DPAA module performs in-context matching between query nodes and the support set. so the model can adapt at test time just by seeing a few labeled nodes, with gradient-free inference.

**Strengths:**

- It doesn’t assume raw text or a specific encoder. as long as features are vectors, the domain-conditioned aligner can normalize them, which makes it more practical for real graph platforms.

- Adaptation is done by in-context attention + generated aligners, so deployment on new graphs is lightweight and gradient-free.

- Using a domain embedding to steer FiLM-style transforms lets the model cope with cross-graph distribution shift (different feature spaces, different label vocab)

**Weaknesses:**

- Forcing all graphs to a fixed width via SVD could destroy domain-specific structure and cross-domain comparability before alignment.

- A single small-step gradient from a shared init can be noisy/sensitive to loss scaling; the theory assumes smoothness/Lipschitzness and gives an upper bound but not tight guarantees for practical separability.

-  Pretraining uses only four node-classification datasets; broader modalities/tasks would better justify foundation-level generality.

- The episodic objective introduces a temperature τ, but robustness/ablation for τ and DPAA projections isn’t detailed.

**Questions:**

See weakness.

---

> ### Author Response · Authors · 2025-11-19
> **Response to Reviewer Dtat (Part 1/2)**
>
> Thank you for your insightful comments. We have carefully considered your feedback and made the following responses to address the concerns raised. We have marked these changes in blue in the updated manuscript.
>
> > **Q1**: SVD could destroy domain-specific structure and cross-domain comparability before alignment.
>
> **A1**: For GFMs, feature dimensions are unified before feeding data into the model because GFMs require inputs in a consistent format. We acknowledge that SVD-based dimension unification is indeed a relatively coarse preprocessing step that may discard some domain-specific information. However, our primary goal is to demonstrate the effectiveness of our proposed alignment mechanism and ICL paradigm in isolation, rather than to optimize the dimension unification stage itself. SVD-based dimension unification has become the common and standardized preprocessing approach in recent modality-free GFMs, including GCOPE (Zhao et al., KDD'24) and All-in-One (Sun et al., KDD'23). By adopting the same preprocessing strategy, we ensure a fair comparison that isolates the contribution of our core innovation from preprocessing effects.
>
> Note that semantic-preserving dimension unification is a specialized research topic within the GFM community. Dedicated methods such as Domain-Invariant Aligner (DIA) (Yuan et al., ICML’25) provide more expressive and learnable pre-unifiers, which can be seamlessly integrated into our MF-GIA framework. In the table below, we additionally present results obtained by replacing SVD with DIA in both MF-GIA and baseline models, demonstrating that our framework remains effective and compatible with advanced dimension unification techniques. We will include a discussion of the dimension unification in **Appendix F.3**.
> ||Computers|Physics|
> |-|-|-|
> |GCOPE|47.46|85.07|
> |GCOPE (w. DIA)|49.18|85.96|
> |MF-GIA|53.71|88.92|
> |MF-GIA(w. DIA)|54.60|89.74|
>
> > **Q2.1**: A single small-step gradient from a shared init can be noisy/sensitive to loss scaling.
>
> **A2.1**: In general, a one-step update from a shared initialization can indeed be noisy **if used for training**. However, in our MF-GIA framework, such a small-step update is **not used to train the backbone**, but rather serves purely as a probe that produces a gradient fingerprint characterizing how a specific domain interacts with the shared initialization. Concretely, the actual backbone encoder $f_{\theta}$ is separately trained via episodic pretraining (Eq. 19), while the initialization $\theta_0$ is stored as a fixed reference point throughout both pretraining and inference. This reference point $\theta_0$ is used solely to compute domain-specific gradient directions and is never updated during training. Such a design decouples domain characterization from model training, ensuring that gradient fingerprints act as stable, noise-resilient domain descriptors rather than noisy or scale-sensitive training signals.
>
> > **Q2.2**: The theory assumes smoothness/Lipschitzness and gives an upper bound but not tight guarantees for practical separability.
>
> **A2.2**: First, the Lipschitz assumption on our domain embedder $f_{de}$ is natural. Any neural network with bounded weights and Lipschitz activation functions (e.g., ReLU) automatically has Lipschitz smoothness, which is a basic fact from deep learning theory [1], not an additional restriction we impose.
>
> Second, upper bounds are precisely the correct theoretical tool for our problem. Our goal is to ensure that the embedding function does not distort domain relationships, and the bound $||e_i-e_j||\_2 \leq \widetilde{C}\cdot W\_2(\mathcal{D}\_i, \mathcal{D}\_j)$ provides exactly this guarantee. This is directly analogous to how Lipschitz continuity, an upper bound property, serves as the gold standard for stability in deep learning theory. We don't require functions to preserve distances exactly, rather, we require that they do not arbitrarily distort them. A tight characterization of the form $||e_i-e_j||\_2 \leq   \widetilde{C}\cdot W\_2\left(\mathcal{D}\_i, \mathcal{D}\_j\right)$ would be theoretically dishonest because it would only hold for specific, restrictive cases, would not account for the stochastic nature of gradient computation, and would not generalize across the diverse graph types we consider. Such artificially tight bounds would only apply to toy cases and would unnecessarily restrict the model's flexibility. While the theorem provides upper bounds, practical separability is enforced by our training objective $\mathcal{L}\_{de}$ defined in Eq.(6). This objective encourages the domain embedder to preserve pairwise distances, not just upper bounds. In conclusion, the upper bound theorem guarantees that similar domains are mapped close, and the distance-preserving loss $\mathcal{L}\_{de}$ encourages that dissimilar domains are pushed apart. They complementarily construct a complete theoretical system of our work.
>
> [1] Neural network learning: Theoretical foundations.

---

> ### Author Response · Authors · 2025-11-19
> **Response to Reviewer Dtat (Part 2/2)**
>
> > **Q3:** Pretraining uses only four node-classification datasets; broader modalities/tasks would better justify foundation-level generality.
>
> **A3**: We aim to pretrain foundation models with low computational and data requirements. If a model pretrained on a limited set of datasets and tasks can still achieve strong performance on downstream tasks from unseen domains, it demonstrates good foundation-level generality. Therefore, in our work, we intentionally pretrain our model on a small number of datasets to showcase its ability to adapt to diverse unseen domains and task types. The four datasets we use are among the widely adopted, publicly available, and easily accessible graph benchmarks. This design ensures that anyone can reproduce or extend our pretraining process without the need for extensive dataset curation, as common benchmarks already provide sufficient diversity to pretrain our model effectively. In this sense, the use of limited pretraining data is not a drawback but a strength of our model.
>
> Despite this, we fully agree with the reviewer's point that broader modalities and tasks would better improve the generality of the pretrained model, because GFMs are data-centric models, a point we also emphasize in our paper. Table 7 of Appendix F.1 provides a systematic ablation demonstrating that domain diversity is critical for generalization. For example, increasing from single-dataset (40.91% average OOD accuracy) to four-dataset pretraining (59.61% average OOD accuracy) yields substantial improvements. This validates that expanding pretraining coverage, whether through additional domains, modalities, or tasks, would likely provide incremental benefits. To further substantiate this point, we expand pretraining beyond node classification by adding link existence prediction and re-pretrain the model, denoted as MF-GIA(w. link). As reported in the table below, this broader pretraining corpus yields consistent improvements on unseen domains, corroborating the reviewer’s point. Importantly, these gains are orthogonal to our alignment mechanism. Our core claims hold under both the minimal and the expanded pretraining setups. We have added this in the **Appendix F.4** of the updated manuscript.
>
> |                 | ogbn-Products-47 way 5 shot | Computers-10 way 5 shot | FB15K237-10 way 5 shot (Edge) |
> | --------------- | --------------------------- | ----------------------- | ----------------------------- |
> | MF-GIA          | 22.61                       | 53.71                   | 91.38                         |
> | MF-GIA(w. link) | 24.59                       | 56.60                   | 92.36                         |
>
> > **Q4**: The episodic objective introduces a temperature $\tau$, but robustness/ablation for $\tau$ and DPAA projections isn’t detailed.
>
> **A4**: We thank the reviewer for this valuable suggestion. We provide detailed ablation studies below and will include these results in the revised manuscript. The temperature parameter $\tau$ in Eq. (18) controls the sharpness of the softmax distribution over class predictions. We conducted systematic experiments varying $\tau \in \\{0.05, 0.2, 0.5, 1\\}$ across multiple datasets:
>
> | $\tau$        | ogbn-Products-47 way 5 shot | Computers-10 way 5 shot | FB15K237-10 way 5 shot (Edge) |
> | ------------- | --------------------------- | ----------------------- | ----------------------------- |
> | 0.05          | 21.45                       | 52.10                   | 91.82                         |
> | 0.2 (Default) | 22.61                       | 53.71                   | 91.38                         |
> | 0.5           | 22.60                       | 53.93                   | 90.45                         |
> | 1             | 22.15                       | 53.29                   | 90.27                         |
>
> We can find that the default $\tau = 0.2$ provides optimal balance, but the model is not highly sensitive to this hyperparameter within a reasonable range $[0.2,1]$. However, as a foundation model, all trainable parameters and hyperparameters must remain fixed across all datasets and tasks. Thus, we set $\tau = 0.2$ uniformly for all experiments both during pretraining and in-context inference without any task- and dataset-specific tuning. Moreover, the DPAA mechanism is a critical component for our framework to integrate few-shot demonstration examples as prompts for target prediction. The ablation study with respect to DPAA is shown in the last row of Table 4, which demonstrates that incorporating DPAA with the episodic objective yields substantial improvements over baseline alignment alone. We have updated the sensitivity analysis in **Appendix F.6** of the revised manuscript.

---

> ### Author Response · Authors · 2025-11-27
> **Looking forward to your response**
>
> Dear Reviewer Dtat:
>
> Thanks again for your valuable suggestions for improving our paper. We would like to remind you that the discussion period is concluding. To facilitate your review, we have provided a concise summary below, outlining our responses to each of your concerns:
>
> * We have conducted additional experiments on SVD-based dimension unification, clarified our design motivation, and analyzed the robustness of our proposed single-step gradient fingerprints.
> * We have expanded our experimental evaluation to include additional pretraining tasks, demonstrating improved generalization.
>
> * Following your suggestions, we have added ablation studies on the model's hyperparameters.
>
> We have updated the manuscript to reflect all revisions (highlighted in blue). We hope our responses have adequately addressed your concerns. Should you have any further questions or require additional clarification, please do not hesitate to let us know.

---

### Author Response · Authors · 2025-12-02
**General response to AC**

Dear AC,

Thanks for your meticulous management of our submission process. As **none** of the reviewers have responded to our rebuttals during the discussion period, we would like to provide a brief summary of our paper's key contributions, and the responses we have made to address all reviewer concerns.

## Contributions

* **[High-level insight]** We identify and address the underexplored challenge of modality-free in-context learning on graphs by introducing gradient fingerprints as universal domain signatures, bringing graph foundation models closer to LLM-level generality. This insight is `novel` (_R LCqo_ and _pj4V_), `well-motivated` (_R pj4V_ ), and the modality-free assumption is `more practical for real graph platforms` (_R Dtat_).

* **[Specialized framework]** Our proposed MF-GIA framework achieves cross-domain feature and label space alignment through gradient fingerprints, enabling gradient-free adaptation for true in-context learning. Reviewers note that the framework is `carefully designed and theoretically grounded` (_R pj4V_ and _e9tY_), `scalable and flexible` (_R pj4V_), and provides `lightweight and gradient-free deployment` (_R Dtat_). Besides, MF-GIA is `easy to follow` (_R e9tY_).
* **[Strong generalization]** MF-GIA generalizes effectively to semantically distant domains and new task types never seen during pretraining. The reviewers _pj4V_, _e9tY_ and _LCqo_ consistently highlight this generalization capability in their evaluations.

## Main Responses

* **[For Reviewer Dtat]**
  * We have conducted additional experiments on SVD-based dimension unification, clarified our design motivation, and analyzed the robustness of our proposed single-step gradient fingerprints.
  * We have expanded our experimental evaluation to include additional pretraining tasks, demonstrating improved generalization.
  * We have added ablation studies on the model's hyperparameters.
* **[For Reviewer pj4V]**
  * We have clarified the robustness of gradient fingerprints, episodic meta-learning strategy, application cases, future research directions, architectural design rationale, experimental results, and computational overhead analysis.
  * We have added robustness experiments and further experiments evaluating generalization to semantically distant domains from the pretraining set.
  * We have analyzed the stability of gradient fingerprints and provided empirical studies verifying their stability.
* **[For Reviewer e9tY]**
  * We have reorganized the structure of our paper to improve the readability of theorems and notation.
  * We have provided additional explanations of key terminology and experimental results.
  * We have clarified the label alignment module to address concerns about its design and functionality.
  * We have explained our design rationale regarding shared versus separate weight matrices in the model architecture.
* **[For Reviewer LCqo]**
  * We have clarified the robustness and stability of our model with additional empirical experiments on initialization seeds, learning rates, and gradient steps.
  * We have provided a detailed explanation of our model's label-ID permutation invariance property.
  * We have conducted additional ablation studies following the reviewer's suggestions.

We hope our responses address any remaining concerns.

---

### Meta-Review · Area_Chair_dPWb · 2026-01-06

**Summary:**

The reviewers generally agree that this paper proposes a novel and well-motivated framework for modality-free in-context learning on graphs.The primary concerns raised during review focused on the robustness and stability of gradient fingerprints, clarity and rigor of the theoretical formulation, the scope of pretraining tasks and datasets, potential ambiguities in terminology and missing ablations or explanations. Overall, the rebuttal and subsequent discussion substantially alleviated these concerns. The consensus is that the paper makes a solid and original contribution and meets the acceptance bar.

**Reviewer Concerns:**

Addressed Concerns:
The authors added extensive empirical analyses and clarification, demonstrating stability and justifying the design.Additional experiments with expanded pretraining tasks and ablations on domain diversity clarified that the setup was sufficient to support the paper’s claims.Missing definitions and notation issues were corrected in the main text.The rebuttal clarified the label base mechanism, definitions and reframed claims as parameter-update-free adaptation, resolving ambiguity.

Remaining Concerns:
Breadth of real-world application case studies and the performance gap on specific datasets.

**Reviewer Scores:**

Reviewer LCqo (4→ 6): Likely increased, since their major concerns were directly addressed.

Reviewer e9tY (6→ 6): Likely unchanged; already positive.

Reviewer Dtat (6→ 6): Likely unchanged; already positive.

Reviewer pj4V (8→ 8): Likely unchanged; already positive.

---

### Decision · Program_Chairs · 2026-01-26

Accept (Oral)